# Transcription factor RonA-driven GlcNAc catabolism is essential for growth, cell wall integrity, and pathogenicity in *Aspergillus fumigatus*

Xiufang Gong,[1,2] Xinwei Ge,[1,3] Qijian Qin,[1] Bin Wang,[1] Linqi Wang,[2] Cheng Jin,[2] Wenxia Fang[1]

**ABSTRACT** *Aspergillus fumigatus,* a saprophytic mold, demonstrates metabolic versatility by utilizing diverse carbon sources to sustain its growth and pathogenic potential. While N-acetylglucosamine (GlcNAc), an ubiquitous amino sugar, serves as a vital nutrient, its catabolic pathway in *A. fumigatus* remains unexplored. Here, we identified core components of this pathway, including GlcNAc-6-phosphate deacetylase (DacA), glucosamine-6-phosphate deaminase (NagA), and the transcription factor RonA. The expressions of *dacA*, *nagA*, and *ronA* were strongly induced when GlcNAc was the sole carbon source. Both ΔdacA and ΔnagA mutants exhibited abolished growth under GlcNAc condition, whereas the ΔronA mutant exhibited pleiotropic defects, including severe growth defects, impaired polarity, delayed development, reduced cell wall integrity, and decreased virulence in a *Galleria mellonella* infection model. The deletion of *ronA* resulted in enhanced immune clearance and exacerbated inflammatory responses. Conidial cell wall analysis revealed that ΔronA conidia displayed aberrant cell wall architecture, primarily characterized by increased surface protein exposure and significantly reduced melanin. Collectively, our findings highlight RonA's critical role in GlcNAc catabolism, conidial cell wall integrity, and the pathogenesis of *A. fumigatus*, providing novel insights into antifungal drug development.

**IMPORTANCE** *Aspergillus fumigatus* is a major human fungal pathogen known for its ability to cause a wide range of diseases, primarily due to its exceptional adaptability to diverse environments. This study identifies DacA and NagA as key enzymes in GlcNAc catabolism, while the transcription factor RonA is essential for growth, sporulation, and cell wall stress response on GlcNAc. Beyond regulating GlcNAc catabolism, RonA was found to play a pivotal role in modifying the conidial cell wall structure, influencing host-pathogen interactions, including immune modulation and pathogenicity. These findings highlight RonA as a potential therapeutic target for treating *A. fumigatus* infections.

**KEYWORDS** *Aspergillus fumigatus*, GlcNAc catabolism pathway, cell wall, virulence, antifungal target

*A*spergillus fumigatus is a saprophytic filamentous fungus and is emphasized as a critical-priority clinical fungal pathogen by the World Health Organization (1–3). Owing to the airborne spores (typically 2 – 3 µm in diameter), *A. fumigatus* disseminates efficiently and is readily inhaled into the human respiratory tract. In immunocompetent hosts, *A. fumigatus* conidia are efficiently cleared by innate immune defenses. However, in immunocompromised individuals, *A. fumigatus* can cause a spectrum of diseases, including invasive aspergillosis (IA), allergic bronchopulmonary aspergillosis, and chronic pulmonary aspergillosis (3–5). Among these, IA is the most lethal

Address correspondence to Wenxia Fang, wfang@gxas.cn, or Cheng Jin, jinc@im.ac.cn.

The authors declare no conflict of interest.

See the funding table on p. 12.

manifestation, representing a leading cause of mortality in intensive care unit patients. Global epidemiological data revealed a dramatic increase in IA cases over the past decade: reported cases surged from over 200,000 annually in 2012 (6) to over 2 million in 2024, with an alarming crude mortality rate exceeding 85% (3). The surging cases underscores the growing public health burden, further exacerbated by the limited antifungal arsenal and rising drug resistance (7–9). Consequently, there is an urgent need for novel therapeutic strategies to combat this life-threatening infection.

N-acetylglucosamine (GlcNAc) is a ubiquitous biomolecule serving as a fundamental structural component of bacterial peptidoglycan, fungal chitin, and the extracellular matrix of mammalian cells (10, 11). Beyond its structural role, GlcNAc also functions as an important signaling molecule across diverse microbial species. In *Escherichia coli*, GlcNAc promotes Curli fibers, essential for biofilm formation (12). In the opportunistic fungal pathogen *Cryptococcus neoformans*, GlcNAc modulates cell wall composition, melanin deposition, and capsule size (13). Notably, in *Candida albicans*, GlcNAc is a well-known inducer of morphological transitions and enhances gastrointestinal colonization (14). Moreover, GlcNAc catabolism contributes to the virulence of various pathogenic fungi, including *C. albicans*, *C. tropicalis*, *Yarrowia lipolytica*, *Histoplasma capsulatum*, *Magnaporthe oryzae*, and *Blastomyces dermatitidis* (14–20).

Interestingly, the model yeasts *Saccharomyces cerevisiae* and *Schizosaccharomyces pombe* lack the genetic machinery to catabolize GlcNAc (21). Extensive studies have elucidated the GlcNAc metabolic pathway in other fungi (10, 22, 23), where extracellular free GlcNAc is imported into cells by the membrane transporter NGT1 and then sequentially metabolized into fructose-6-phosphate (Fru6P)—a central metabolic intermediate—via three stepwise enzymes: hexokinase (HXK1), phosphorylates GlcNAc to form GlcNAc-6-phosphate (GlcNAc6P); deacetylase (DAC1), deacetylates GlcNAc6P to produce glucosamine-6-phosphate (GlcN6P); deaminase (NAG1) converts GlcN6P into ammonium and Fru6P. Fru6P is a key metabolic intermediate having multiple fates in cells (10, 24). Phylogenetic analysis indicates that fungal GlcNAc catabolism is regulated by RON1 (**r**egulator **o**f **N**-acetylglucosamine catabolism 1), a transcription factor harboring an Ntd80-like DNA-binding domain (21). The importance of *ron1* regulation on GlcNAc utilization has been validated in *Trichoderma reesei*, *C. albicans,* and *C. tropicalis* (18, 21, 25).

As a saprophytic pathogen, *A. fumigatus* exhibits remarkable metabolic adaptability, enabling it to thrive in both environmental and host niches. Our previous work demonstrated that *A. fumigatus* efficiently utilizes GlcNAc as a sole carbon source (26), yet the GlcNAc catabolic gene cluster remains undefined in this species. In this study, we identified and functionally characterized key GlcNAc catabolism genes in *A. fumigatus*. We showed that *dacA* (deacetylase) and *nagA* (deaminase) are specialized enzymes for GlcNAc catabolism, while the transcription factor *ronA* exhibits pleiotropic regulatory roles beyond GlcNAc catabolic regulation.

## MATERIALS AND METHODS

### Strains and growth conditions

The *A. fumigatus* strain CEA17 is an uracil auxotrophic strain carrying a point mutation in the *pyrG* gene, resulting in a $pyrG^-$ phenotype (27). Importantly, CEA17 retains the wild-type (WT) *akuB* (also known as *KU80*) gene. A derivative strain lacking *akuB* ( Δ*akuB*) and also *pyrG* was constructed and designated as Δ*akuB*$^{KU80}$/*pyrG*$^-$ (28). This strain is commonly used to facilitate targeted gene deletion due to the increased efficiency of homologous recombination resulting from *akuB* deletion. Yeast glucose medium (YG) was prepared containing 0.5% (wt/vol) yeast extract, 2% (wt/vol) glucose, 50 µL/mL salt solution, and 1 µL/mL trace element solution (pH = 6.5). One liter salt solution was composed of 120 g $Na_3NO_3$, 10.4 g KCl, 30.4 g $KH_2PO_4$, 10.4 g $MgSO_4·7H_2O$, and ultrapure $H_2O$ to 1,000 mL. The trace element (TE) solution was composed of 2.2 g $ZnSO_4·7H_2O$, 1.1 g $H_3BO_4$, 0.5 g $MnCl_2·4H_2O$, 0.5 g $FeSO_4·7H_2O$, 0.16 g $CoCl_2·5H_2O$, 0.16

g $CuSO_4$, 0.11 g $(NH4)_6Mo_7O_{24} \cdot H_2O$, 5 g EDTA, and ultrapure $H_2O$ to 100 mL. YG agar slant was utilized for sporulation, and conidia were collected using 0.2% (vol/vol) Tween 20 and counted by a hemocytometer (29). YGU medium (YG supplemented with 5 mM each of uridine and uracil) was used for the cultivation of the $\Delta akuB^{KU80}/pyrG^-$ strain to generate protoplasts. Minimal medium (MM) was prepared containing 1% (wt/vol) glucose), 50 µL/mL salt solution, and 1 µL/mL TE. Solid plates were prepared by adding 1.5% agar (26).

## Identification of the GlcNAc catabolism pathway

To validate gene expression induced by GlcNAc, $2 \times 10^8$ fresh spores of the WT were pre-cultured in MM at 37°C with shaking at 200 rpm for 24 hours. Then mycelia were aseptically collected and divided into two equals: one continuously inoculated into MM and the other into MMG for GlcNAc induction. After 2 hours of submerged cultivation, mycelia were promptly collected, flash-frozen in liquid nitrogen, and stored at −80°C. RNA extraction, first-strand cDNA biosynthesis, and qRT-PCR were performed to calculate the relative expression of each gene by the $2^{-\Delta\Delta Ct}$ method (30). TBP (*AFUB_039050*) encodes a TATA-binding protein and is used as the reference gene (31, 32).

## Construction of mutant and revertant strains in the GlcNAc catabolic pathway

All primers used in this study are listed in Table S1. Targeted gene (*dacA*, *nagA,* and *ronA*) deletions were generated via homologous recombination as previously described (33, 34). Briefly, ~1 kb flanking fragments of the target gene were infused with the *neo-AnpyrG-neo* cassette. The assembled fragments were subsequently transformed into $\Delta akuB^{KU80}/pyrG^-$ protoplasts. Transformants were selected on MM plates, and single colonies were verified by genomic PCR (35).

For revertant strain (RT) construction, fragment 1 (the upstream flanking region 1,000 bp + the gene + downstream 100 bp) was amplified by F1/R1, fragment 2 (*pyr4* selective marker) was amplified by F2/R2, and fragment 3 (downstream 1000 bp of fragment 1) was amplified by F3/R3, respectively. Three purified fragments were then infused into the pCE-Zero vector to form the revertant (RT) complementation plasmid. This recombinant plasmid served as a template for PCR amplification of the full-length complementation cassette using flanking primers F1/R3 and transformed into $\Delta gene\Delta pyrG$ protoplasts. The resulting amplicon was subsequently purified and transformed into $\Delta gene\Delta pyrG$ protoplasts via polyethylene glycol (PEG)-mediated transformation. The $\Delta gene\Delta pyrG$ strains were screened and acquired on YGU containing 1 mg/mL 5-Fluprppratoc acid (5-FOA). Single colonies from GlcNAc minimal medium plates were picked for genomic DNA extraction and PCR analysis.

### *Southern blotting analysis*

Conidia suspensions were aseptically inoculated into YG liquid medium for genome extraction. After submerged culture at 37°C for 36 h, mycelia were promptly harvested by filtration, flash-frozen in liquid nitrogen, and stored at −80°C until genome extraction. The DNA extraction buffer (50 mL) was prepared as follows: 5 mL 1 M Tris-HCl (pH = 8.0), 0.4 mL 0.5 M EDTA-$Na_2$ (pH = 8.0), 5 mL 20% SDS, 2 mL 5 M NaCl, and ultrapure $H_2O$ to 50 mL. After liquid nitrogen grinding, ~100 mg nitrogen-pulverized mycelium per tube was homogenized in 900 µL DNA extraction buffer, followed by phenol-chloroform extraction twice. For each extraction, tubes were centrifuged at 12,000 rpm × 10 min at 4°C, and the aqueous phase was transferred to a new tube. Nucleic acids were precipitated with three volumes of ice-cold pure ethanol at −20°C for ≥ 30 min, washed twice with 70% ethanol, air-dried, and resuspended in nuclease-free water. DNA integrity was verified by 1% agarose gel electrophoresis, and concentration was quantified using a NanoDrop spectrophotometer.

Five primer pairs were employed to confirm genome manipulation. Primer pair sequences and amplifications were detailed in the schematic diagrams (Fig. S1) and primer table (Table S1).

Southern blotting was performed using the DIG DNA Labeling and Detection Kit (Roche), following the manufacturer's instructions. For each target gene, a ~ 1 kb DIG-labeled probe was designed to hybridize to a sequence downstream of the coding region as indicated in Fig. S1A, D and G. For the probes of Δ*dacA* and Δ*nagA*, the 1 kb probes were right downstream of the gene, respectively, while for Δ*ronA* detection probe, its probe started 100 bp after the gene and extended for 1,000 bp. Genomic DNA from WT, Δ*dacA*, Δ*ronA*, and Δ*ronA* revertant (rRT) strains was digested with *Bgl* II, while Δ*nagA* genomic DNA was digested with *Nco* I. Digestions were carried out at 37 °C for 4 – 6 hours. The digested DNA was separated by 1% agarose gel at 40 V for over 6.5 hours. After electrophoresis, DNA was denatured, neutralized, and transferred onto nitrocellulose membranes via capillary blotting, followed by UV crosslinking (1 J, 2 min). Membranes were prehybridized at 68 °C for 30 minutes and then hybridized with the corresponding DIG-labeled probe. The optimal hybridization temperature was calculated according to the kit's instructions. Stringency washes and signal detection were performed as recommended in the kit protocol.

## Plate assays

For the carbon utilization assays, MM plates were supplemented with various carbon sources (1% wt/vol), including glucose (Glc), glucosamine (GlcN), GlcNAc, fructose, maltose, mannose, arabinose, galactose, xylose, sucrose, glycerol, and ethanol. For sensitivity assays, stressors were added as follows: 50 µg/mL Congo Red (CR), 100 µg/mL Calcofluor White (CFW), 1.2 M sorbitol, 0.8 M NaCl, 0.6 M KCl, 70 µg/mL Hygromycin B (Hph), 50 µg/mL SDS, 5 mM $H_2O_2$, 0.1 µg/mL voriconazole (VOR), 1 µg/mL amphotericin B (AmB), 2 µg/mL caspofungin (CAS), and 256 µg/mL fluconazole (FLU), respectively. Spores ($10^5$–$10^2$) were incubated at 37°C for 40 h before photographs were captured.

## Melanin extraction and conidial cell wall component analysis

Conidial cell wall component analysis was conducted on Glc. After 3 days of culture at 37°C, WT and Δ*ronA* mutant spores were freshly collected and used for cell wall component analysis. Conidial surface-exposed proteins were extracted by 0.5 M NaCl incubation for 2 h at room temperature at a ratio of $10^{10}$ conidia per ml (36). The NaCl supernatant was retrieved by centrifugation, and protein concentration was interpolated from a standard curve using Coomassie Brilliant Blue Solution (Transgene, China). For hydrophobic layer extraction, dry conidia ($10^9$) were incubated in 400 µL formic acid for 2 h on ice (37). After centrifugation, the supernatant was dried in a 40°C vacuum drying oven for 2 h to remove formic acid and dissolved in 100 µL of 10 mM phosphate-buffered saline (PBS; pH = 7.5). Samples (10 µL) of WT and Δ*ronA* were separated by 15% SDS-PAGE and stained by silver staining (Sangon biotech, China). Melanin extraction was conducted using $10^{10}$ spores per tube, followed by treatment with a combination of Vino Taste Pro (Novozymes, China), guanidine thiocyanate (Sangon, China), Proteinase K (Sangon, China), and boiling in 6 M HCl for 1 h (38–40). The melanin ghosts were then vacuumed, dried thoroughly, and weighed (39). Conidia ($10^9$) were ground by the grinder mill for cell wall polysaccharide and protein analysis (41). Quantification of a-/β-glucans followed the phenol/sulfuric acid method (42). Chitin content was determined by measuring the GlcN released after acid hydrolysis (43). Cell wall proteins were interpolated from the Coomassie brilliant blue standard curve. Biofilm assay followed our published method (26). Briefly, $10^5$ conidia of each strain were inoculated in a 96-well plate and cultivated in RPMI 1640 containing glucose. After 48 h, biofilm was measured by crystal violet staining.

## Virulence assay

The virulence test was conducted using *Galleria mellonella* model (33, 44). Briefly, sixth instar larvae were selected and divided into five groups (100 larvae per group). Using a Hamilton syringe, 10 microliter of freshly collected spores ($7 \times 10^7$/ mL) in 0.2% Tween 20 was injected into the hind proleg of each larva. For the control group, the same volume of 0.2% Tween 20 was injected. Survival rates were recorded at 24, 48, 72, and 96 h post-injection. Larvae that were immobile and displayed dark spots or apparent melanization were considered dead.

## *In vitro* internalization assay in A549

The internalization of *A. fumigatus* into the human lung epithelial cell line A549 was analyzed as described (19, 45). Briefly, the A549 cells were cultured in complete DMEM (supplemented with 10% fetal bovine serum [FBS], 0.1% streptomycin, and 0.1% gentamicin). Approximately $10^4$ A549 cells per well were seeded in a 24-well plate overnight. The next day, the culture medium was removed, and fresh fungal spores were added at a multiplicity of infection (MOI) of 5 and incubated at 37°C with 5% $CO_2$ for 6 hours. After washing three times with sterile PBS, 20 µg/mL nystatin was added for an additional 4 h incubation. The monolayer was then washed three times with PBS and lysed with PBS containing 0.25% Triton X-100. The released conidia were plated onto MM plates after appropriate dilution. Single colonies were calculated after 36 h of culture. The internalization rate was calculated as the percentage of conidia colonies relative to the initial inoculum.

## RAW 264.7 cell assays

RAW 264.7 cell assays were cultured in complete DMEM for phagocytosis and survival assays. For the phagocytosis index assay, $10^5$ RAW 264.7 cells/well were seeded into a 24-well plate with a 12 mm sterile cover glass in each well. Spores were labeled by fluorescein isothiocyanate (FITC; 0.48 µg/mL) at 37°C for 1 h, and the unlabeled dye was removed by PBS washing and centrifugation. FITC-labeled spores were added at an MOI of 5. The plate was centrifuged at 120 *g* × 5 min to synchronize phagocytosis. Two hours later, the plate was washed twice with sterile PBS and dyed with Dil (3 µM, Beyotime, China) at 37°C for 10 min. PBS washing for three times, then the cells were fixed in 4% paraformaldehyde at 37°C for 20 min. The cover glasses were put upside down on the glass slides for phagocytic index calculation by the Keyence microscope.

Survival rates within RAW 264.7 cells were also done by the nystatin protection assay as described for internalization. Briefly, $10^4$ cells/well were seeded in a 48-well plate and infected by $5 \times 10^4$ spores. After 2 h incubation, the medium was discarded and 20 µg/mL nystatin was added for a further 4 h of incubation. Spores were acquired by cell lysis and spread on MM plates by suitable dilution. Single colonies were calculated after incubation at 37°C for 36 h.

Cytotoxicity assay was conducted by measuring the lactate dehydrogenase (LDH) leakage, following the manufacturer's instructions (CK12, Dojindo). Briefly, 5,000 RAW 264.7 cells per well were seeded in DMEM (containing 5% FBS), and spores were added at an MOI of 2. After 24 h incubation, 10 µL lysis buffer was added to the high control and incubated at 37°C 5% $CO_2$ for 30 min. Then 100 µL working solution was added into each well and cultured for 30 min. Absorbance measurement at $OD_{490\ nm}$ was performed immediately after adding the stop solution. Cytotoxicity was calculated by comparing the absorbances of the WT group or Δ*ronA* group to the high control.

For the TNF-α secretion assay, the RAW 264.7 cells were seeded in FBS-free DMEM in a 24-well plate ($1 \times 10^6$ cells/well). Freshly harvested spores were added at MOI of 1 and 5. Supernatants were collected after 10 hours of co-culture at 37°C in a $CO_2$ incubator. The supernatants were used immediately or stored at −20°C until detection. TNF-α concentration was determined by interpolation from the mouse TNF-α ELISA standard curve (Jiu Bang Sheng Wu, China).

The qRT-PCR assays were carried out to detect the expression levels of TNF-α, IL-1, IL-1β, IL-6, MCP-1, IL-12, and CCL2. Briefly, $2 \times 10^6$ RAW 264.7 cells were seeded in FBS-free DMEM in a 24-well plate and then infected with WT or ΔronA conidia (MOI = 5) for 12 h. After washing in PBS once, the plate was flash-frozen in liquid nitrogen and stored at −80°C. RAW 264.7 cells were used for RNA qRT-PCR performance as mentioned above. Gene expression was quantitatively normalized against β-actin. All primers are listed in Table S1.

## Statistical analysis

The Kaplan-Meier survival curves were plotted using GraphPad Prism 8, and statistical significance for survival differences was assessed with the log-rank (Mantel-Cox) test. For comparisons between two groups, a *t*-test was used. For multiple comparisons, a one-way ANOVA multiple-comparison test was performed to determine statistical significance.

## RESULTS

### Identification of putative GlcNAc catabolic genes in *A. fumigatus*

Extensive investigation of the *C. albicans* GlcNAc catabolic pathway has established a foundation for homologous identification in other fungal species (Fig. 1A) (14, 17, 18, 22, 23, 46–49). By BLAST analysis using *C. albicans* DAC1 (AJW76789.1), NAG1 (AAA34352.1), and RON1 (AOW30785.1) as queries against *A. fumigatus* genome, we identified functional homologs, DacA, NagA, and RonA, in both A1163 and Af293. Notably, Af293 contains two copies of these genes while A1163 possesses a single set (Table 1), with all genes being non-clustered in the genome in A1163 (Fig. 1B).

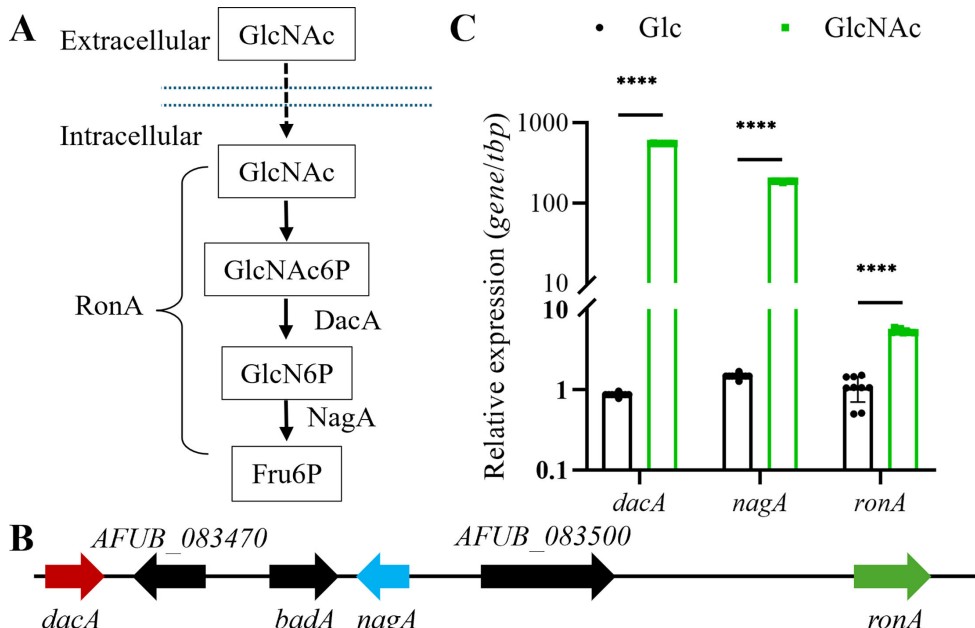

**FIG 1** Putative GlcNAc catabolism components in *A. fumigatus*. (A) Blasted with *C. albicans*, schematic representation of the GlcNAc catabolic pathway in *A. fumigatus*, DacA and NagA are catalytic enzymes regulated by RonA. (B) Genomic organization of the predicted *dacA*, *nagA*, and *ronA* and the adjacent genes in *A. fumigatus* A1163. AFUB_083470 encodes a GMC family oxidoreductase, AFUB_083480 encodes a betaine aldehyde dehydrogenase (*badA*), and AFUB_083500 encodes a beta-N-acetylglucosaminidase. (C) qRT-PCR analysis showing transcript expression of the GlcNAc catabolic pathway upon GlcNAc induction. Data represent means ± standard deviations (SD) (*n* = 9) from independent experimental replicates. Asterisks denote statistically significant differences (***$P < 0.0001$).

**TABLE 1** Putative GlcNAc metabolic homologs in *A. fumigatus* by tBLASTn.

| Gene ID in *A. fumigatus* | | *C. albicans* | Identity | Similarity | Function |
|---|---|---|---|---|---|
| A1163 | Af293 | | | | |
| *AFUB_083460* | *AFUA_8 G04100/* *AFUA_1G00450* | AJW76789.1 | 38.81% | 56.67% | GlcNAc-6-P deacetylase, DacA |
| *AFUB_083490* | *AFUA_8 G04070/* *AFUA_1G00480* | AAA34352.1 | 53.18% | 68.64% | GlcN-6-P deaminase, NagA |
| *AFUB_083510* | *AFUA_8 G04050/* *AFUA_1G00580* | AOW30785.1 | 21.52% | 43.05% | PacG/VIB-1 Ndt80 family, RonA |

As the GlcNAc catabolic pathway is known to be strictly induced by GlcNAc (16, 21, 22, 26), we performed GlcNAc induction of *A. fumigatus* mycelium for 2 h followed by qRT-PCR analysis. This showed *dacA*, *nagA*, and *ronA* were upregulated by 554.2-fold, 187.2-fold, and 5.3-fold (Fig. 1C), respectively.

## The GlcNAc catabolic pathway specializes in amino sugar utilization

To explore the physiological role of the GlcNAc catabolic pathway, we generated and verified deletion mutants of Δ*dacA,* D*nagA,* and Δ*ronA*, and the respective revertant strains (dRT, nRT, and rRT). Genotypic confirmation was achieved through PCR amplification with five primer pairs and Southern blot (Fig. S1).

To verify whether *dacA*, *nagA,* and *ronA* are specifically tailored for amino sugar utilization in *A. fumigatus*, we performed spot assays on solid plates containing either GlcN or GlcNAc as the sole carbon source, with glucose (Glc) as a control.

On GlcN plates, the Δ*dacA* mutant and dRT strain exhibited growth comparable to WT, whereas the Δ*nagA* mutant showed complete growth inhibition, and Δ*ronA* displayed significant growth defects (Fig. 2). This observation is consistent with the metabolic flux, where GlcN is phosphorylated to GlcN-6-phosphate (GlcN6P), subsequently converted to fructose-6-phosphate (Fru6P) before entering glycolysis and other metabolic pathways (Fig. 1A) (50).

On GlcNAc plates, both Δ*dacA* and Δ*nagA* mutants failed to grow, while Δ*ronA* showed severe growth inhibition, forming sparse, transparent hyphae rather than the opaque colonies typical of WT strains.

These findings corroborate previous studies in *C. albicans* and verified *dacA*, *nagA,* and *ronA* are essential for amino sugar catabolism in *A. fumigatus* (25). Unexpectedly, we noticed that the three mutants, Δ*dacA*, Δ*nagA*, and Δ*ronA*, showed growth defects on alternative carbon sources, such as Glc (Fig. 2), while the revertant strains cannot fully

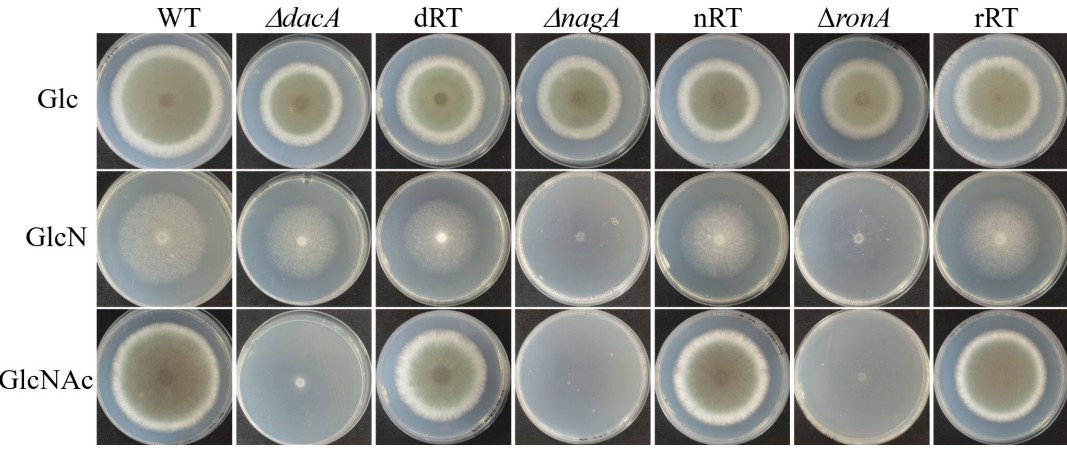

**FIG 2** The GlcNAc catabolism pathway is specialized to use amino sugars. Spores ($10^4$) of each strain were spotted on plates containing only Glucose, GlcN, or GlcNAc. Plates were photographed after 96 h incubation at 37°C.

restore to the WT (Fig. S2), indicating metabolic changes or regulatory imbalances that persist after gene manipulation.

## RonA affects the cell wall integrity on GlcNAc

GlcNAc is the constitutional unit of chitin, which localizes in the inner layer of the cell wall and contributes to the cell wall rigidity of the *A. fumigatus* (51). Defects in GlcNAc catabolism might lead to chitin reduction and then affect the entire cell wall integrity. To validate this hypothesis, we tested the sensitivities of Δ*dacA*, Δ*nagA*, and Δ*ronA* mutants to cell wall-disrupting agents (CFW and CR), cell membrane-perturbing agent (SDS), osmotic agents (sorbitol, NaCl, and KCl), protein biosynthesis inhibitors (Hph), antifungal drugs targeting the cell membrane (VOR, FLU, and AMB), and cell wall (CAS).

Stress tolerance assays were performed under both Glc and GlcNAc conditions. On MM, no significant growth differences were observed between the mutants and WT strains under various stress conditions (Fig. 3A). However, we noticed that Δ*ronA* was more resistant to hph, indicating deletion of *ronA* altered the cellular protein profile.

When cultured on GlcNAc-containing minimal medium, both Δ*dacA* and Δ*nagA* abolished growth. However, supplementation with sorbitol partially restored the growth defects in these mutants, likely through its conversion to Fru1P serving as an alternative carbon source for *A. fumigatus* (52). The Δ*ronA* mutant demonstrated increased sensitivity to Congo red (CR) and calcofluor white (CFW), consistent with compromised cell wall integrity (Fig. 3B). Intriguingly, all three osmotic stress agents, as well as caspofungin (CAS) (Fig. S3), restored the Δ*ronA* mutant growth to the WT levels, although the precise regulation mechanism remains to be elucidated.

Further phenotypic characterization revealed that *ronA* deletion resulted in hypersensitivity to hph, indicative of impaired protein biosynthesis in *A. fumigatus* on GlcNAc, while maintaining normal sensitivity to SDS, $H_2O_2$, azoles, and amphotericin B (AMB) (Fig. 3B; Fig. S3). Collectively, these findings demonstrate that *ronA* plays an essential role in maintaining cell wall integrity during growth on GlcNAc (Fig. 3B).

## Deletion of *ronA* led to increased proteins and decreased melanin in the conidia cell wall

During spore collection from the YG slant and subsequent spot assays, we noticed that the Δ*ronA* spores exhibited obvious lighter pigmentation compared to the WT strain, indicating a reduction in melanin content. Given that melanin is critical for maintaining

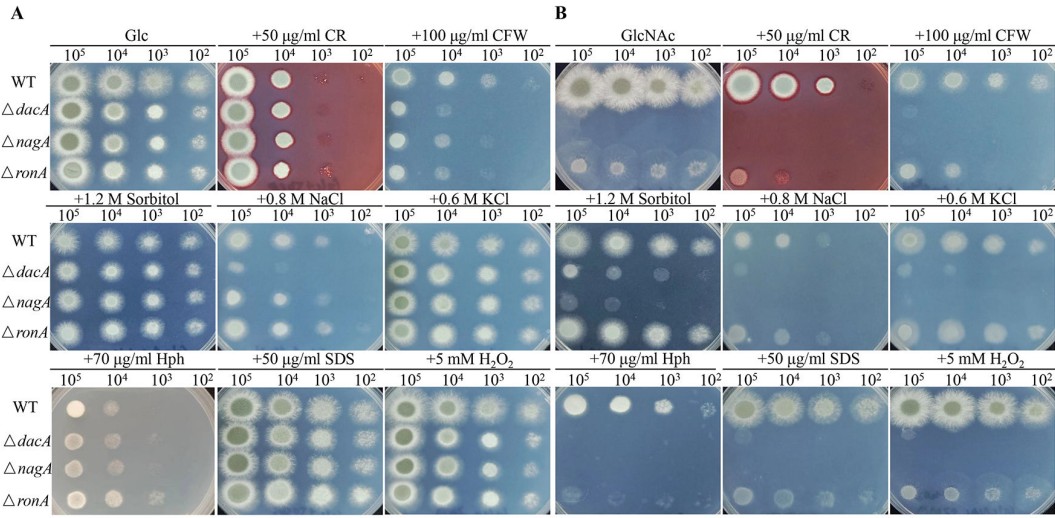

**FIG 3** Sensitivity assays of GlcNAc catabolic mutants to chemical agents on Glc (A）and GlcNAc (B). The stressors added to each plate were indicated above. Serial dilutions ($10^5$–$10^2$ spores) of each strain were inoculated and incubated at 37°C for 48 hours before photographing.

the conidial cell wall rigidity and structural integrity (53, 54), we proceeded to analyze the conidial cell wall composition of ΔronA spores.

Quantitative assays confirmed that melanin levels were significantly reduced by 84.16% in the ΔronA mutant strain (0.67 ± 0.21 mg in the mutant versus 4.233 ± 0.775 mg in the WT, $P < 0.01$) (Fig. 4A and B). In contrast, conidial surface-deposited proteins increased by 145.27% (10.89 ± 0.65 in the mutant versus 4.44 ± 0.15 µg in the WT, $P < 0.0001$) (Fig. 4C), while cell wall glycoproteins increased by 32% (178.28 ± 28.75 µg the mutant versus 134.87 ± 14.22 µg in the WT, $P < 0.001$) (Fig. 4D). Particularly, the hydrophobin RodA increased abundance in ΔronA resting spores as indicated by silver staining (Fig. 4E). However, no significant differences were detected in α-/β-glucans, chitin, or biofilm formation (Fig. 4D and F). Taken together, these findings demonstrate that ronA deletion perturbs conidial cell wall composition, leading to reduced melanin deposition and increased conidial proteins.

## Deletion of *ronA* attenuated virulence and increased inflammatory response

Evaluation of the GlcNAc catabolic pathway in fungal virulence using *G. mellonella* infection model (35, 55). While ΔdacA and ΔnagA mutants phenocopied WT virulence profile (Fig. S4), the ΔronA mutant demonstrated significantly attenuated virulence, with a survival rate of 49% at 96 hours post-injection ($P < 0.0001$) (Fig. 5A; Table S2).

Given the role of lung epithelial cells in initial spore clearance, we performed internalization assays using A549 cells (56, 57). The ΔronA mutant exhibited significantly impaired host cell invasion, with recovery rates of only 2.3% ± 0.9 compared to 18.6% ± 3.7 for WT spores ($P < 0.01$) (Fig. 5B). This reduced infectivity was further evidenced in macrophage interactions, where ΔronA spores showed a 56.1% decrease in survival

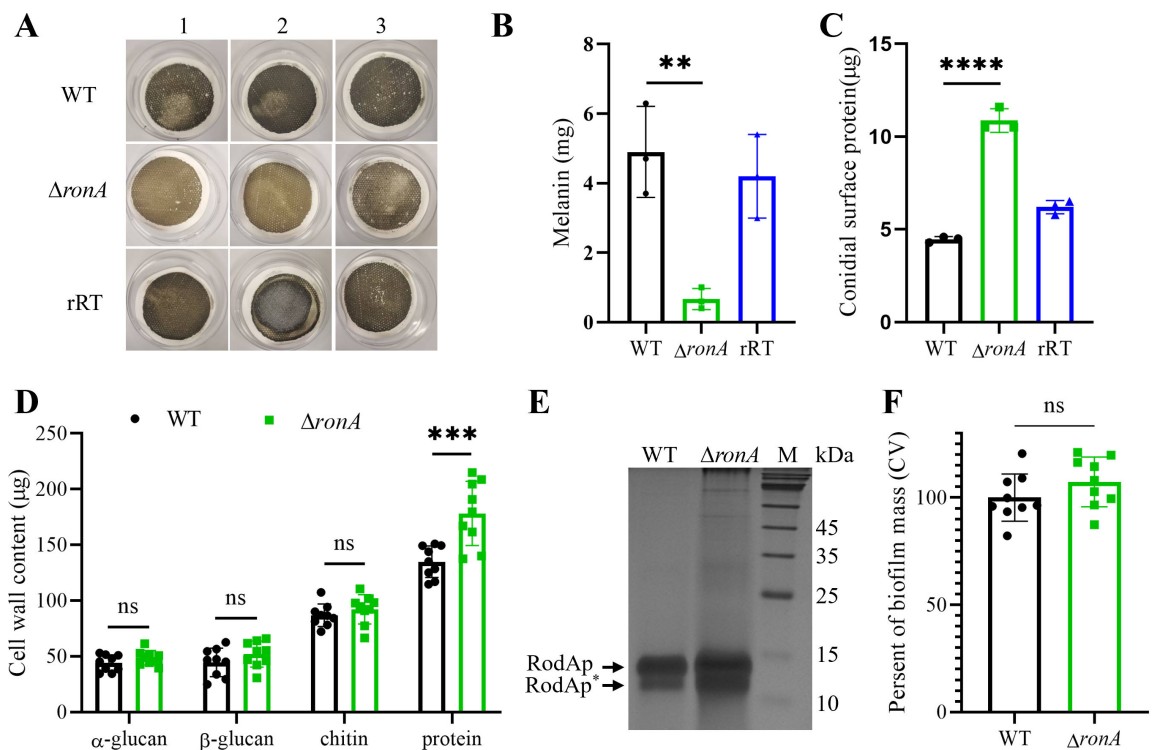

**FIG 4** Conidial cell wall component comparison of the WT strain and ΔronA strain. (A) Visualization of melanin and (B) weight comparison of melanin. (C) Quantification of conidial surface-deposited proteins extracted by 0.5 M NaCl, $10^{10}$ dry spores were used for each strain. (D) Contents of α-/β-glucans, chitin, and cell wall-related glycoproteins extracted from $10^9$ fresh conidia. (E) SDS-PAGE profile of the hydrophobic layer extracted from formic acid. The bands were visualized by silver staining. RodAp* was degraded from the RodA caused by formic acid treatment. Data are presented as means ± SD from independent experimental replicates. (F) Biofilm formation quantification by growing the WT strain and ΔronA strain in the complete DMEM medium (ns, $P > 0.05$; *$P < 0.05$; ***$P < 0.001$; ****$P < 0.0001$).

within RAW 264.7 macrophages (Fig. 5C), despite comparable phagocytosis rates (Fig. 5D).

Cytotoxicity assessment revealed Δ*ronA* spores induced 58.9% less LDH release than WT (Fig. 5E), indicating diminished host cell damage. Paradoxically, this hypovirulent strain elicited significantly elevated TNF-α production at both MOI 5 and 1 after 10 hours incubation (Fig. 5F). Transcriptional profiling confirmed this hyperinflammatory response, with qRT-PCR demonstrating upregulation of IL-6, IL-1β, MCP-1, IL-12, and CCL2 (Fig. S5).

These results demonstrate that RonA-mediated GlcNAc catabolism significantly impacts fungal pathogenesis through both direct infectivity and immunomodulatory mechanisms.

## DISCUSSION

Opportunistic fungal pathogens like *A. fumigatus* have evolved sophisticated nutrient acquisition systems that contribute to their ecological success and pathogenicity. A critical aspect of this adaptation is their metabolic flexibility in utilizing diverse carbon sources, enabling survival in challenging host environments (58). While this metabolic versatility is well-recognized, the specific roles of individual catabolic pathways in *A. fumigatus*-host interactions remain poorly understood.

During infection, *N*-acetylglucosamine (GlcNAc) becomes available through degradation of host structural components, including glycosaminoglycans, glycoproteins, and proteoglycans (58). Beyond its nutritional value, GlcNAc metabolism has

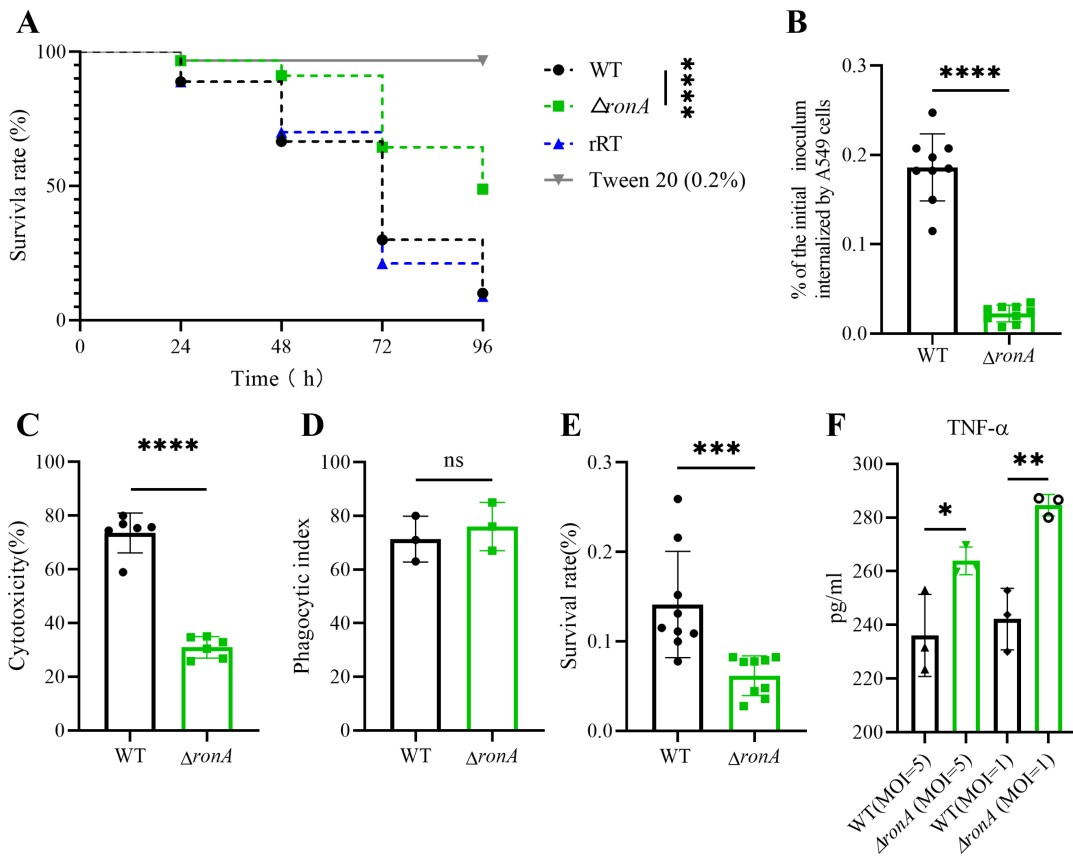

**FIG 5** Deletion of *ronA* attenuates virulence and triggers inflammatory responses. (A) Kaplan-Meier curves of the survival rates of *G. mellonolla* larvae at 24, 48, 72, and 96 h post-conidial injection. Three biological replicates were conducted for each strain. (B) Internalization of spores in A549 cells and (D) Survival rates of spores in RAW 264.7 cells were assessed using the nystatin protection assay. (C) The phagocytosis index represents the number of phagocytosed spores per 100 RAW 264.7 cells. (E) LDH releases from infected cells, expressed as a proportion of the high-control group after 24 h of infection. (F) TNF-α secretion by RAW 264.7 cells after infection with WT and Δ*ronA* spores for 10 h. All data are presented as means ± SD from independent experimental replicates (ns, no significance; *P < 0.05; **P < 0.01; ***P < 0.001; ****P < 0.0001).

established itself as a virulence determinant in multiple pathogens, including *C. albicans, C. deneoformans,* and *Leishmania major* (47, 59).

Our comparative genomic analysis identified three conserved GlcNAc catabolic genes in *A. fumigatus: dacA*, *nagA*, and *ronA*. This coordinated and significant induction indicates that these genes are involved in GlcNAc catabolism. Interestingly, there was one set in A1163 and 2 sets in Af293. Af293 is a clinical isolate obtained in 1993 from a lung biopsy (60), while A1163 is a derivative of the clinical isolate CEA10 (27). Phylogenetic analysis of 169 *A. fumigatus* genomes classified Af293 in cluster III and A1163 in cluster I (61). Although both strains have eight chromosomes and over 97% gene identity, significant genetic variation exists (62). This genetic diversity explains why Af293 harbors two copies of the GlcNAc metabolic genes, whereas A1163 contains only one set. Further functional studies are planned to determine which gene copies in Af293 are responsible for GlcNAc catabolism.

Phenotypic characterization revealed that Δ*dacA* and Δ*nagA* abolished GlcNAc utilization, reconfirming their roles as specialized enzymes for GlcNAc catabolism (Fig. 2) (21, 25, 63). However, the Δ*ronA* mutant retained partial growth capacity on GlcNAc (Fig. 2), constructing with the observations in *C. albicans*, *C. tropicalis,* and *T. reesei* (18, 21, 25). This species-specific difference suggests alternative regulatory mechanisms for GlcNAc metabolism in *A. fumigatus*. Interestingly, osmotic stress and CAS exposure restored Δ*ronA* growth to WT levels (Fig. 3A and B; Fig. S3), implicating potential crosstalk between the HOG pathway (high osmolarity glycerol) (64) and CWI (cell wall integrity) pathway (65) and GlcNAc utilization, which warrants further investigation.

The Δ*dacA* and Δ*nagA* mutants exhibited WT virulence difference in our *G. mellonella* infection model (Fig. S4), contrasting with the attenuated phenotypes reported for corresponding mutants in *C. albicans* and *M. oryzae* (24, 47). This species-specific discrepancy may stem from distinct nutritional strategies during infection. While GlcNAc serves as a primary carbon source for other pathogens (14, 66), *A. fumigatus* demonstrates remarkable metabolic flexibility, capable of utilizing alternative carbon sources present in the host environment. Consequently, the growth defects observed in Δ*dacA* and Δ*nagA* mutants were strictly limited to GlcNAc as the sole carbon source, with no measurable impact on pathogenicity in nutrient-complex environments (Fig. 2; Fig. S2).

In contrast, the Δ*ronA* mutant exhibited significantly attenuated virulence despite maintaining growth capacity on non-amino sugar carbon sources. Comprehensive phenotypic analysis revealed impaired survival within immune cells, enhanced TNF-α production by host immune cells, and upregulation of inflammatory cytokines and chemokines (Fig. 5; Fig. S5). These findings suggest that *ronA* regulates *A. fumigatus* virulence via structuring modifications of conidia. Quantitative analysis demonstrated a 145.27% increase in surface-exposed proteins in Δ*ronA* spores (Fig. 4C). In WT conidia, the hydrophobin RodA confers hydrophobicity and shields the fungus from immune recognition (67, 68). The dramatic increase in surface protein exposure in Δ*ronA* mutants likely uncovers pathogen-associated molecular patterns, thereby potentiating accelerated immune responses (36, 54, 69). This phenomenon parallels observations in melanin-deficient mutants (Δ*pksP*, Δ*ayg1*, and Δ*arp2*), where exposed glycoproteins enhance dendritic cell activation (54), and in Δ*ags1*/D*ags2*/D*ags3* mutants displaying immediate immune recognition due to exposed glycoprotein matrices (36).

Although the specific proteins exposed in Δ*ronA* mutant spores remain to be identified, their increase likely accounts for both heightened immune response and reduced virulence (70). Additionally, the observed reduction in melanin content in Δ*ronA* mutant spores may also contribute to their virulence defect (Fig. 4A and B). In *A. fumigatus,* melanin provides critical protection by (i) mitigating phagolysosomal acidification, (ii) scavenging reactive oxygen species (ROS), (iii) modulating neutrophil recruitment, and (iv) regulating chemokine secretion via post-translational mechanisms (40, 71–73). Future studies will aim to identify the specific proteins exposed on Δ*ronA* mutant spores and elucidate the precise mechanisms driving their enhanced immune-mediated clearance.

In conclusion, this study identified and characterized the GlcNAc catabolic pathway in *A. fumigatus*, revealing distinct functional roles for the three essential components. Our findings demonstrated that *dacA* and *nagA* encode essential enzymes specifically required for GlcNAc catabolism, and *ronA* functions as the critical GlcNAc catabolism regulator, maintains proper cell wall integrity, and modulates host-pathogen interaction. The pleiotropic nature of RonA-mediated regulation creates a unique vulnerability in fungal pathogenesis, suggesting RonA as a promising target for novel antifungal strategies. Its dual role in metabolic regulation and cell wall biogenesis suggests that pharmacological inhibition could simultaneously impair fungal nutrition and enhance host immune detection. This work advances our understanding of fungal pathogenesis by elucidating how metabolic adaptation interfaces with immune evasion strategies in *A. fumigatus*.

## ACKNOWLEDGMENTS

This work was supported by the Guangxi Natural Science Foundation (2023GXNSFFA026011) to W.F., and the National Natural Science Foundation of China (32371339) to W.F., the Guangxi Natural Science Foundation (2025GXNSFBA069549) to X.G., and the Basic research fund of Guangxi Academy of Sciences (2024YWF2111) to X.G.

W.F. conceived the study. X.G., X.G., and Q.Q. performed the experiments. X.G., C.J., and W.F. analyzed and interpreted the data and wrote the manuscript with input from all authors. All authors have read and agreed on the final manuscript.

## AUTHOR AFFILIATIONS

[1]Institute of Biological Sciences and Technology, Guangxi Academy of Sciences, Nanning, Guangxi, China
[2]State Key Laboratory of Microbial Diversity and Innovative Utilization, Institute of Microbiology, Chinese Academy of Sciences, Beijing, China
[3]College of Life Science and Technology, Guangxi University, Nanning, Guangxi, China

## AUTHOR ORCIDs

Xiufang Gong http://orcid.org/0009-0003-1471-4813
Linqi Wang http://orcid.org/0000-0002-5243-341X
Cheng Jin http://orcid.org/0000-0002-1514-6374
Wenxia Fang http://orcid.org/0000-0001-8055-2072

## FUNDING

| Funder | Grant(s) | Author(s) |
| --- | --- | --- |
| Natural Science Foundation of Guangxi Zhuang Autonomous Region | 2020GXNSFDA238008, 2023GXNSFFA026011 | Wenxia Fang |
| Natural Science Foundation of Guangxi Zhuang Autonomous Region | 2025GXNSFBA069549 | Xiufang Gong |
| National Natural Science Foundation of China | 32371339 | Wenxia Fang |
| Basic research fund of Guangxi Adademy of Sciences | 2024YWF2111 | Xiufang Gong |

## AUTHOR CONTRIBUTIONS

Xiufang Gong, Conceptualization, Investigation, Methodology, Writing – original draft | Xinwei Ge, Data curation, Investigation, Methodology | Bin Wang, Conceptualization | Linqi Wang, Supervision, Writing – review and editing | Cheng Jin, Conceptualization, Supervision, Writing – review and editing | Wenxia Fang, Conceptualization, Supervision, Writing – review and editing.

## ADDITIONAL FILES

The following material is available online.

### Supplemental Material

**Supplemental figures (Spectrum00122-25-s0001.docx).** Figures S1 to S5.
**Table S1 (Spectrum00122-25-s0002.xlsx).** Primers used in this study.
**Table S2 (Spectrum00122-25-s0003.xlsx).** Survival rates of *G. mellonella* in each group.

### Open Peer Review

**PEER REVIEW HISTORY (review-history.pdf).** An accounting of the reviewer comments and feedback.

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
