## [Reviewer comments · Microbiology Spectrum]

Microbiology Spectrum

Transcription Factor RON1-Driven GlcNAc Catabolism Is Essential for Growth, Cell Wall Integrity, and Pathogenicity in *Aspergillus fumigatus*

Xiufang Gong, Xinwei Ge, Qijian Qin, Bin Wang, Linqi Wang, Cheng Jin, and Wenxia FANG

Corresponding Author(s): Wenxia FANG, Guangxi Academy of Sciences

Review Timeline:

Submission Date:	January 12, 2025
Editorial Decision:	February 23, 2025
Revision Received:	May 24, 2025
Editorial Decision:	June 29, 2025
Revision Received:	July 24, 2025
Editorial Decision:	August 1, 2025
Revision Received:	August 25, 2025
Accepted:	September 11, 2025

Editor: Miguel Penalva

Reviewer(s): The reviewers have opted to remain anonymous.

Transaction Report:

DOI: <https://doi.org/10.1128/spectrum.00122-25>

Re: Spectrum00122-25 (Transcription Factor RON1-Driven GlcNAc Catabolism Is Essential for Growth, Cell Wall Integrity, and Pathogenicity in *Aspergillus fumigatus*)

Dear Prof. Wenxia FANG:

I have now received two excellent reports on your submission. As you may see from the attached reviews, both referees concur in that your manuscript requires very substantial modification before being accepted for publication. In addition to a number of modifications, some of them very substantial, requested by the referees, the criticisms can be summarised as follows: the conclusions require controls that are currently missing, such as verification of the integration events and the construction of complementation events; second, phenotypic assessment of strains is often incomplete; if you are willing to submit a revised version, please pay special attention to addressing all these concerns, as I will need to return the paper to the same referees to confirm that you have dealt their concerns in a satisfactory way.

Revision Guidelines

Sincerely,
Miguel Penalva
Editor
Microbiology Spectrum

Reviewer #1 (Comments for the Author):

In this work, Gong and coworkers characterize functionally three proteins of *Aspergillus fumigatus* necessary in N-

acetylglucosamine metabolism, with special focus on the transcription factor ron1. The authors analyzed expression levels of these three genes, the phenotypes in multiple culture media for their null mutants, germination patterns, conidia production, cell-wall and conidial outer layer composition, and virulence.

In my opinion, the work is of interest for the readers of mSpectrum, but it needs improvement. First, please, check writing (e.g., in the title, replace "drived" by "driven"). Second, the authors describe that the null mutants generated only show clear phenotypes when grown on GlcNAc or GlcN, but I see growth and conidiation phenotypes on other media, such as when xylose or mannose are used as carbon sources. How do the authors explain that? I would recommend quantification of radial growth and conidia production, not with all inocula tested in Figure 2 but when 1000 or 10,000 conidia are inoculated. Third, some of the quantification criteria used by the authors should be more consistent. For example, in lines 155-156, "The internalization rate was calculated as the percentage of conidia colonies relative to initial inoculum." Is this a reliable method for quantification? How do the authors guarantee that these colonies on MM plates developed from single and isolated conidia?. And finally, I saw no complementation or overexpression strain for the genes analyzed in this work. Wouldn't these strains be necessary? I attach to my report a pdf copy of the manuscript including all my comments and suggestions. Hope it will help.

Reviewer #2 (Comments for the Author):

In the paper by Gong et al. is afforded the study of GlcNAc catabolism and its relevance in virulence of *A. fumigatus* based on previous studied on other fungal pathogens. Although this study focuses on a topic well known in other pathogens, it is worthy to ascertain whether GlcNAc catabolism in *A. fumigatus* functions like in other fungal pathogens or it has some distinctive feature that somewhat enhances the significance of this process in the virulence of *A. fumigatus*. However, this paper has major weaknesses that should be addressed before publication:

- 1) Although mutants were constructed in a KU80 genetic background, they should be analyzed by southern blot to preclude ectopic integration of transformant DNA in any other loci. In addition, reconstituted strains should be generated for each mutant as the best controls for all observed phenotypic traits.
- 2) Lines 278-295. Testing the effect of different stressing compounds (except sorbitol and salts) on fungal mutant strains grown in the presence of GlcNAc as sole carbon source does not make sense since these mutants do not grow under these culture conditions. They should be better tested in the presence of glucose as sole carbon source (as shown for antifungals in Fig.S2). In addition, most phenotypic descriptions are not accurate leading to misinterpretations. For example, in line 282-3 it is stated that "The Δ ron1 mutant was more sensitive to CR and CFW, indicating the cell wall integrity was impaired", but this is not true if considered that wild-type strain also grows worse in the presence of CR and CFW than in their absence. The effect of sorbitol and salts, which could protect fungal strains against osmotic lysis in hypotonic media, argue in favor of a role of Ron1 in maintaining cell wall integrity. However, the statement in line 291 "ron1 is crucial in maintaining cell wall integrity" is an overinterpretation of results. To show that this is as stated, the strains should be grown with sorbitol or salts in the presence of a carbon source other than GlcNAc to tests whether the level of GlcNAc synthesized intracellularly and used for chitin biosynthesis is regulated by Ron1.
- 3) Figure 1B does not reflect the actual arrangement of genes *dac1*, *nag1* and *ron1* in the *A. fumigatus* genome and the alphanumeric codes are wrong (also in table 1): AFUB_083510 is *ron1* (not *dac1*) and AFUB_08346 is *dac1* (not *ron1*). Also, the schematic representation of the locus shown in figure 1B suggests erroneously that these genes are in a cluster. Although they are closely located one each other in the genome, they are separated by other genes. Hence, the figure 1B should be rebuilt including the all genes between them and including a scale.

Other points to be addressed are the following:

- 1) Line 199. Similarities or identities? Inconsistence with Table 1? It should be indicated both. Indicate also the alphanumeric codes for these genes in the reference fungal strain Af293. In case there were any noticeably difference between A1163 and Af293 regarding these genes, it should be discussed. Genes should not be named as *dac1*, *nag1* and *ron1*. Instead, they should be named as *dacA*, *nagA* and *ronA* throughout the text according to rule most largely accepted for gene nomenclature in filamentous fungi.
- 2) Line 80-81. The notation of the strain CEA17 genotype is badly written (is not Δ ku80 Δ pyrG-). CEA17 is an auxotrophic for Uracil (PyrG-) because it has a substitution (not a deletion) in a nucleotide of the *pyrG* gene and it carries the wild-type *akuB* (KU80) gene. The CEA17 derived Δ akuB and PyrG- fungal strain was created by Silva Ferreira et al. (2006, Eukaryotic Cell 5:207-211) and named as KU80 Δ pyrG (reference not cited in the text).
- 3) Line 115. Growth curves for filamentous fungi growing in liquid media cannot be measured by OD (during 5 days!!!).
- 4) Line 157-188. Indicate the culture media used for all these experiments.
- 5) Line 180. After washing with PBS?
- 6) Line 174. What means high control?
- 7) Line 200. In figure 1B is not show any comparison but the catabolic cluster for GlcNAc.
- 8) Line 202 and figure 1C. It is not clear at all how expression levels were relativized. What means gene/tbp?
- 9) Line 221 and from now on. Is GlcN or GlcN6P?
- 10) Line 222. For me the expression "impaired growth" is the same that "growth was completely inhibited". Figure 2 should be cited at this point.
- 11) Line 224. The Δ ron1 mutant does not act as a transcription factor (the transcription factor is Ron1). You should pay more attention in wording properly.
- 12) Lines 236-273. This is worthless and could be deleted.

- 13) Line 278. Could this explain the difference in the grow between wild-type and each mutant with glucose, fructose, mannose, sucrose and xylose as sole carbon sources as seen in figure 2?
- 14) Lines 329-330. Not consistent with data shown in Fig. 6B.

DRIVEN

**Transcription Factor RON1-Driven GlcNAc Catabolism Is Essential for Growth,**
**Cell Wall Integrity, and Pathogenicity in *Aspergillus fumigatus***

Xiufang Gong^{1,2#}, Ge Xinwei^{1#}, Qijian Qin¹, Bin Wang¹, Linqi Wang², Cheng Jin^{1,2*} and Wenxia
Fang^{1*}

¹Institute of Biological Sciences and Technology, Guangxi Academy of Sciences, Nanning,
Guangxi, China.

²State Key Laboratory of Mycology, Institute of Microbiology, Chinese Academy of Sciences,
Beijing, China

*Corresponding author's email: wfang@gxas.cn or jinc@im.ac.cn

#These authors contributed equally to this work.

**Abstract**

*Aspergillus fumigatus*, a saprophytic mold, utilizes diverse carbon sources to support its growth
and pathogenicity. Among these, N-acetylglucosamine (GlcNAc), an abundant amino sugar, serves
as a vital nutrient. However, the GlcNAc catabolic pathway in *A. fumigatus* remains unexplored.
This study identifies key components of this pathway, including GlcNAc-6-phosphate deacetylase
(DAC1), glucosamine-6-phosphate deaminase (NAG1), and the transcription factor RON1. The
expression of *dac1*, *nag1*, and *ron1* was strongly induced when GlcNAc was the sole carbon
source. Mutants lacking *dac1* or *nag1* exhibited abolished growth under GlcNAc conditions, while
the *ron1* mutant showed severe growth defects, impaired polarity, delayed development, reduced
cell wall integrity, and decreased virulence in a *Galleria mellonella* infection model. The *ron1*
mutant also displayed enhanced immune clearance and a **heightened** inflammatory response.
Conidial cell wall analysis revealed increased surface protein exposure and significantly reduced
melanin in *ron1* mutants. These findings highlight RON1's critical role in GlcNAc catabolism,
conidial cell wall integrity, and **the** pathogenesis of *A. fumigatus*.

**Importance**

*Aspergillus fumigatus* is a major human fungal pathogen known for its ability to cause a wide
range of diseases, primarily due to its exceptional adaptability to diverse environments. This study
identifies DAC1 and NAG1 as key enzymes in GlcNAc catabolism, while the transcription factor
RON1 is essential for growth, sporulation, and cell wall stress response **on GlcNAc**. Beyond
regulating GlcNAc catabolism, RON1 was found to play a pivotal role in **modifying** the conidial
cell wall structure, influencing host-pathogen interactions, including immune modulation and
pathogenicity. These findings highlight RON1 as a potential therapeutic target for treating *A.*
*fumigatus* infections.

Introduction

*Aspergillus fumigatus* is a saprotrophic filamentous fungus and listed as a first prioritized clinical
fungal pathogen by the World Health Organization (1). Owing to the small size of its airborne
spores, *A. fumigatus* can be easily inhaled into the human respiratory tract. While healthy
individuals typically eliminate these inhaled spores without difficulty, the conidia can cause a
broad range of diseases in immunocompromised individuals, including invasive aspergillosis (IA),
allergic bronchopulmonary aspergillosis (ABPA), and chronic pulmonary aspergillosis (CPA) (2,
3). IA, in particular, is the most common infectious cause of death among ICU patients. In 2012,
over 200,000 cases of IA were reported annually (4), and by 2024, this number had surged to over
2,000,000 cases, resulting in an annual crude mortality rate of over 85.2% (5). The increasing
prevalence of IA poses a significant threat to public health, highlighting the urgent need for the
development of novel drugs due to the limited availability of effective treatments (6-8).

N-acetylglucosamine (GlcNAc) is a ubiquitous molecule found in nearly all living organisms.
It is well known as a structural component of bacterial cell wall peptidoglycan, fungal cell wall
chitin, and the extracellular matrix of human cells (9, 10). Besides its structural role, GlcNAc also
functions as an important signaling molecule. Curli fibers, promoted by GlcNAc, are crucial for
biofilm formation in *Escherichia coli* (11). In addition, GlcNAc influences cell-wall composition,
melanin architecture, and capsule size in *Cryptococcus neoformans* (12). Notably, GlcNAc is a
well-known inducer of morphological changes and gastrointestinal colonization in *Candida*
*albicans* (13). Moreover, GlcNAc catabolism plays a role in the virulence of various pathogenic
fungi, including *C. albicans*, *C. tropicalis*, *Yarrowia lipolytica*, *Histoplasma capsulatum*,
*Magnaporthe oryzae*, and *Blastomyces dermatitidis* (13-19).

The model organisms *Saccharomyces cerevisiae* and *Schizosaccharomyces pombe* lack the
necessary genes to catabolize GlcNAc (20). Extensive studies on GlcNAc pathway have been
particularly significant for understanding its function in various fungi (9, 21, 22). Extracellular
free GlcNAc is transported into cells by the membrane transporter NGT1 and then catabolized into
fructose-6-phosphate (Fru6P) via three stepwise enzymes: hexokinase (HXK1), phosphorylating
GlcNAc to form GlcNAc-6-phosphate (GlcNAc6P); deacetylase (DAC1), deacetylating
GlcNAc6P to produce glucosamine-6-phosphate (GlcN6P); deaminase (NAG1), converting
GlcN6P into ammonium and Fru6P. Fru6P is a key metabolic intermediate having multiple fates

in cells (9, 23). Phylogenetic analysis of the GlcNAc gene cluster has shown that the GlcNAc
catabolism pathway is regulated by RON1 (regulator of N-acetylglucosamine catabolism 1) with a
Ntd80-like domain (20). The importance of *ron1* regulation on GlcNAc utilization has been
demonstrated in *Trichoderma reesei*, *C. albicans* and *C. tropicalis* (17, 20, 24).

As a saprotrophic pathogen, *A. fumigatus* has evolved to adapt to and thrive on both
environmental and host niches. Our previous research demonstrates that *A. fumigatus* grows well
on GlcNAc as the sole carbon source, indicating the fungus can effectively utilize GlcNAc (25).
However, the widely studied GlcNAc catabolic gene cluster remains undefined in *A. fumigatus*. In
this study, we identified key GlcNAc catabolism genes in *A. fumigatus*. Function analysis revealed
*dac1* and *nag1* were specialized enzymes in GlcNAc catalyzation, the transcription factor *ron1*
showed multiple roles beyond GlcNAc catabolic regulation.

**Materials and Methods**

**Strains and growth conditions**

The *A. fumigatus* strain CEA17 ($\Delta Ku80\Delta pyrG^-$) was used as the parental strain for mutant
construction, while the CEA17C ($\Delta Ku80pyrG^+$) strain was utilized as the wild type (WT) for
phenotype analysis. YG solid medium was utilized for sporulation, and conidia were collected by
using 0.2% (vol/vol) Tween 20 (26). YGU (adding 5 mM uridine and uracil to YG) was used for
cultivation of the CEA17 strain to generate protoplasts. Minimal Medium (MM) and MMG
(replacing glucose by GlcNAc) were prepared as described and solid plates were prepared by
adding 1.5% agar (25).

**Identification of the GlcNAc catabolism pathway**

To validate gene expression induced by GlcNAc, 2×10^8 /ml fresh spores of the WT were pre-
cultured in MM at 37 °C with shaking at 200 rpm for 24 hours. Then mycelia were aseptically
collected and divided into two equals: one continuously inoculated into MM and the other into
MMG for GlcNAc induction. After two hours' submerged cultivation, mycelia were promptly
collected, flash-frozen in liquid nitrogen, and stored at -80 °C. RNA extraction, first-strand cDNA
biosynthesis and qRT-PCR were performed as described in our previous publication (27).

**Mutant construction of the GlcNAc catabolic pathway**

Primers employed for mutant construction and verification were enlisted in Supplementary Table
1. The $\Delta dac1$, $\Delta nag1$ and $\Delta ron1$ mutants were constructed using homologous recombination as
previously described (28, 29). Briefly, approximately 1000 bp homologous fragments of target
gene were infused with the *neo-AnpyrG-neo* cassette. The assembled fragments were subsequently
transformed into CEA17 protoplasts. Single colonies from MM plates were picked for genomic
DNA extraction and PCR analysis (30).

**Plate assays**

For the carbon utilization assays, MM plates were supplemented with various carbon sources,
including glucose, GlcNAc, glucosamine (GlcN), fructose, maltose, mannose, arabinose,
galactose, xylose, glycerol, sucrose, and ethanol. For sensitivity assays, stressors were added as
follow: 50 μ g/ml Congo Red (CR), 100 μ g/ml Calcofluor White (CFW), 1.2 M sorbitol, 0.8 M
NaCl, 0.6 M KCl, 70 μ g/ml Hygromycin B (Hph), 50 μ g/ml SDS, 5 mM H₂O₂, 0.1 μ g/ml

voriconazole (VOR), 1 $\mu\text{g/ml}$ amphotericin B (AmB), 2 $\mu\text{g/ml}$ caspofungin (CAS) and 256 $\mu\text{g/ml}$
fluconazole (FLU), respectively. Spores (10^5 to 10^2) were incubated at 37 °C for 40 hours before
photographs were captured.

**Germination, growth and conidiation assays**

Germination depiction was conducted in MM and MMG submerged cultures. Ten hours post-
culture, 100 cells of each strain were photographed for germination rate analysis. Radial growth
assay was performed on MM or MMG plates, 10^5 spores of each *A. fumigatus* strain were
inoculated, and the colony diameters were gauged every 24 hours for 5 days. In parallel, growth
curves of liquid MMG culture were monitored at OD_{530 nm} every 12 hours for 5 days. After 7 days'
culture on MMG plates, spores were collected and numerated for conidiation analysis.

[revised manuscript text omitted]

homologous **genes** were retrieved, showing 39%, 53.2 % and 24.41% amino acid similarities
(Table 1, Figure 1B), respectively. The GlcNAc catabolic pathway is strictly induced by GlcNAc
(15, 20, 21, 25), therefore, *A. fumigatus* mycelium was induced by GlcNAc for 2 h. The qRT-PCR
showed *dac1*, *nag1*, and *ron1* were upregulated by 554.2-fold, 187.2-fold, and 5.3-fold (Figure
1C), respectively. This coordinated and significant upregulation indicates that the gene cluster is
involved in GlcNAc catabolism. To explore the physiological role of the GlcNAc catabolic
pathway, $\Delta dac1$, $\Delta nag1$, and $\Delta ron1$ mutants were generated by homologous recombination
through the *neo-AnpyrG-neo* selection cassette flanked by **upstream and downstream** fragments,
and mutants were verified by PCR with multiple pairs of primers (Supplementary Figure 1).

Table 1. Putative GlcNAc metabolic cluster in *A. fumigatus* by tBLASTn.

Gene ID in A. fumigatus	C. albicans	Identity	Function
AFUB_083510	AJW76789.1	39%	GlcNAc-6-P deacetylase, DAC1
AFUB_083490	AAA34352.1	52.37%	GlcN-6-P deaminase, NAG1
AFUB_083460	CR04250WA	23.43%	PacG/VIB-1 Ndt80 family, RON1

Figure 1. Putative GlcNAc catabolism components in *A. fumigatus*. (A) Schematic representation
of the GlcNAc catabolic pathway, including key enzymes: NGT1, HXK1, DAC1, and NAG1. This
pathway is regulated by RON1. (B) Genomic organization of the predicted GlcNAc catabolic
homologs in *A. fumigatus*. (C) qRT-PCR analysis showing transcript expression of the GlcNAc
**catabolic pathway upon GlcNAc induction**. Data represent means \pm standard deviations (SD) (n =
9) from independent experimental replicates. Asterisks denote statistically significant differences
(****, $P < 0.0001$).

**The GlcNAc catabolic pathway is specialized for amino sugar utilization**

To verify that **the GlcNAc catabolism homologues** are specifically tailored for GlcNAc utilization

in *A. fumigatus*, sole carbon source spot assay was performed. As anticipated, on MMG plate, both
the $\Delta dac1$ and $\Delta nag1$ mutants exhibited abolished growth. On plates with GlcN, the $\Delta dac1$ mutant
showed impaired growth, while the $\Delta nag1$ mutant's growth was completely inhibited, this was due
to its incapability of converting GlcN6P into Fru6P, which is an important metabolite for
glycolysis and other metabolic pathways (Figure 1A) (45). The $\Delta ron1$ mutant, which acts as a
transcription factor, presented significantly reduced growth on GlcNAc and even more
pronounced growth inhibition on GlcN (Figure 2), indicating it is an amino sugar utilization
regulator (24). All the three mutants did not show obvious growth defects on the other carbon
plates (Figure 2). These findings suggest that the *A. fumigatus* GlcNAc catabolic pathway is
specifically dedicated to amino sugar utilization.

Figure 2. Growth of the GlcNAc catabolism gene cluster mutants on amino sugars and other
carbon sources. Diluted spores (10^5 - 10^2) of WT, $\Delta dac1$, $\Delta nag1$, and $\Delta ron1$ strains were grown on
minimal medium plates containing sole carbon source. Plates were photographed after 48 h
incubation at 37 °C.

**Deletion of *ron1* resulted in reduced cell growth, decreased conidiation, and disordered**
**germination in the presence of GlcNAc**

Since $\Delta dac1$ and $\Delta nag1$ mutants showed no growth on GlcNAc as the sole carbon source, only the
WT strain and $\Delta ron1$ mutant strain were employed for the growth rate assay. A total of 10^5 spores
from each strain were inoculated at the center of MMG plates and cultured for 5 days. Although
the radial growth of the $\Delta ron1$ mutant was nearly identical to that of the WT, the hyphae of the
$\Delta ron1$ mutant were conspicuously thin to transparent (Figure 3A), indicating defective growth in
the absence of RON1 when GlcNAc is the sole carbon source. This observation was further
confirmed by comparing the growth rates in liquid MMG. After 2 days, the $OD_{530\text{ nm}}$ of the WT
strain exceeded 0.77 ± 0.03 , while the $OD_{530\text{ nm}}$ of the $\Delta ron1$ mutant was only 0.09 ± 0.01 (Figure
3B).

For conidiation measurement on GlcNAc, after the growth rate assay, the spores were then
collected and counted using a hemocytometer. The WT strain exhibited normal sporulation, while

no spores were collected from the $\Delta ron1$ mutant (Figure 3C), indicating that sporulation in the
$\Delta ron1$ mutant was completely suppressed.

When examining germination rates in the early stages on solid MM plates, normal
germination rates were observed in all *A. fumigatus* strains. However, on solid MMG plates,
neither the $\Delta dac1$ nor $\Delta nag1$ mutants exhibited germination, reconfirming their inability to grow
when GlcNAc is the sole carbon source. After 10 hours' incubation at 37 °C, the WT strain
produced a single long germ tube, while the $\Delta ron1$ mutant exhibited two shorter germ tubes, often
forming angles less than 180 degrees, and in some cases, less than 90 degrees, indicating
disordered polarity establishment in the $\Delta ron1$ mutant (Figure 3D). Germination rate analysis
revealed a 52% reduction in the $\Delta ron1$ mutant (99 ± 1 in the WT vs 48 ± 8 in the $\Delta ron1$ mutant)
(Figure 3D, Table 2), illustrating that germination rate is significantly reduced in the $\Delta ron1$
mutant. Taken together, these results demonstrate that the GlcNAc cluster is essential for the
survival of *A. fumigatus* when GlcNAc is the sole carbon source.

Figure 3. Comparison of growth rate, sporulation, and germination between WT and GlcNAc
catabolic mutants. (A) Radial growth of WT and $\Delta ron1$ strains on solid MMG plates. A total of
10^5 spores from each strain were inoculated onto MMG solid plates and incubated at 37 °C.
Photos were taken on day 3 and day 5 post-incubation. (B) Growth comparison between WT and
$\Delta ron1$ mutant strains in MMG submerged culture. A total of 10^5 spores from each strain were
inoculated for 48 hours, with OD_{530 nm} absorbance measurements taken every 8 hours. (****, $P <$
0.0001). (C) Conidia production was quantified using a hemocytometer after 7 days' incubation on
MMG at 37 °C. (D) Germination morphology of the GlcNAc catabolism pathway mutants in
liquid MM and MMG. Photos were taken at 10 h. Scale bar, 25 μ m. Data are presented as means \pm
SD from independent experimental replicates (****, $P < 0.0001$).

Table 2. Germination rates of the GlcNAc catabolic mutants.

Medium	WT	$\Delta dac1$	$\Delta nag1$	$\Delta ron1$
MM	99 ± 1	99 ± 1	99 ± 1	99 ± 1
MMG	99 ± 1	0	0	48 ± 8

**GlcNAc catabolic pathway affects the cell wall integrity**

GlcNAc is the constitutional unit of chitin, which localizes in the inner layer of cell wall and
contributes to the cell wall rigidity of **the** *A. fumigatus* (46). Defects in GlcNAc catabolism might
lead to **chitin reduction** and then affect entire cell wall integrity. To validate this hypothesis, we
tested the sensitivities of **GlcNAc catabolic mutants** to cell wall disrupting agents (CFW and CR),
cell membrane perturbing agent (SDS), the **osmotic agents** (sorbitol, NaCl and KCl), protein
biosynthesis inhibitor (Hph), antifungal drugs targeting the cell membrane (VOR, FLU and
AMB), and cell wall (CAS). As expected, the $\Delta ron1$ mutant was more sensitive to CR and CFW,
indicating the cell wall integrity was impaired (Figure 4). **Although no growth of $\Delta dac1$ and**
**$\Delta nag1$ mutants on GlcNAc,** sorbitol partially restored the growth of both $\Delta dac1$ and $\Delta nag1$
mutants. Probably because sorbitol is a sugar alcohol which can be metabolized into Fru1P and
acts as a carbon source to support the growth of *A. fumigatus* (47). Intriguingly, all the three
osmotic stress agents and CAS (Supplementary Figure 2) restored the growth of $\Delta ron1$ **mutant to**
**the WT,** although the regulation mechanism underlying is unknown (Figure 4). **Deletion of *ron1***
**was more sensitive to** Hph, indicating that **the** protein biosynthesis was inhibited in *A. fumigatus*,
**while no sensitivity to SDS, H₂O₂, azoles or AMB** (Figure 4 and Supplementary Figure 2). Taken
together, *ron1* is crucial in maintaining cell wall integrity (Figure 4).

Figure 4. Sensitivity assays **of GlcNAc catabolic mutants to chemical agents** on MMG. The
stressors added to each plate were indicated above. Serial dilutions (10^5 - 10^2 spores) of each strain
were inoculated and incubated at 37 °C for 48 hours before photographing.

**Deletion of *ron1* led to increased proteins and decreased melanin in conidia cell wall**

When performing spore collection from YG and spot assays, we noticed that the pigment of $\Delta ron1$
spores was lighter than the WT strain, indicating **the** reduction of melanin in $\Delta ron1$ conidia. Since
**the** melanin is important for structuration and stiffness of the conidial cell wall (48, 49), we thus
determined **to analyze** the conidial cell wall components. As expected, the amount of melanin of
**the $\Delta ron1$ spores** was indeed reduced by 26% (1.67 ± 0.21 mg in $\Delta ron1$ mutant strain *versus* 2.27
± 0.23 mg in the WT strain, $P < 0.05$) (Figure 5A and 5B), while conidial surface-deposited

proteins increased by 188% ($14.90 \pm 0.57 \mu\text{g}$ in $\Delta ron1$ mutant strain versus $5.16 \pm 0.65 \mu\text{g}$ in the
WT strain, $P < 0.0001$) (Figure 5C), and cell wall glycoproteins increased by 32% (178.28 ± 28.75
306 μg in $\Delta ron1$ mutant strain versus $134.87 \pm 14.22 \mu\text{g}$ in the WT strain, $P < 0.001$) (Figure 5D). Cell
wall polysaccharides showed no difference in α -/ β -glucans, chitin or biofilm (Figure 5D and 5E).
In addition, the RodA protein also increased in $\Delta ron1$ mutant rest spores as indicated by silver
staining (Figure 5F). Taken together, deletion of *ron1* led to increased conidial surface-deposited
proteins and reduced melanin.

Figure 5. Conidial cell wall component comparison of the WT strain and $\Delta ron1$ strain. (A)
Visualization of melanin and (B) weight comparison of melanin. (C) Quantification of conidial
surface-deposited proteins extracted by 0.5 M NaCl, 10^{10} dry spores were used for each strain. (D)
Contents of α -/ β -glucans, chitin and cell wall-related glycoproteins extracted from 10^9 fresh
conidia. (E) Biofilm formation quantification by growing WT strain and $\Delta ron1$ strain in the
complete DMEM medium. (F) SDS-PAGE profile of the hydrophobic layer extracted from formic
acid. The bands were visualized by silver staining. RodAp*, degraded from the RodA caused by
formic acid treatment. Data are presented as means \pm SD from independent experimental
replicates (ns, $P > 0.05$; *, $P < 0.05$; ***, $P < 0.001$; ****, $P < 0.0001$).

***ron1* deletion attenuated virulence and increased inflammatory response**

To evaluate the role of the GlcNAc catabolic pathway in virulence, a *G. mellonella* infection
model was employed. The results showed that the $\Delta dac1$ and $\Delta nag1$ mutants exhibited no
significant changes in virulence. In contrast, the $\Delta ron1$ mutant demonstrated significantly
attenuated virulence, with a survival rate of 28% at 96 hours post-injection ($P < 0.001$) (Figure 6A
and Supplementary Table 2).

Lung epithelial cells, as the first line of defense against *A. fumigatus* spore invasion, play a
critical role in spore clearance (50, 51). Internalization assay demonstrated that single colonies of
the $\Delta ron1$ mutant retrieved from A549 cells were less than 87% of the WT strain (Figure 6B).
Consistent with this finding, the survival rate of the $\Delta ron1$ mutant in RAW 264.7 macrophages
decreased by 56.1%, although the phagocytosis index remained unchanged (Figure 6C and 6D).

Furthermore, the $\Delta ron1$ mutant exhibited less cytotoxicity to host cells, as indicated by a 58.9%
reduction in lactate dehydrogenase (LDH) release (Figure 6E). Notably, the $\Delta ron1$ mutant induced
higher levels of TNF- α , regardless of the multiplicity of infection (MOI) being 5 or 1 after 10
336 hours' incubation (Figure 6F). The qRT-PCR analysis revealed upregulation of pro-inflammatory
cytokines and chemokines, including IL-6, IL-1 β , MCP-1, IL-12, and CCL2, suggesting an
enhanced inflammatory response to $\Delta ron1$ mutant spore infection (Supplementary Figure 3). In
summary, deletion of $\Delta ron1$ leads to attenuated virulence while triggering a stronger inflammatory
response.

Figure 6. Deletion of *ron1* attenuates virulence and triggers inflammatory responses. (A) Kaplan-
Meier curves of the survival rates of *G. mellonella* larvae at 24, 48, 72, and 96 h post-conidial
injection. Three biological replicates were conducted for each strain. (B) Internalization of spores
in A549 cells and (D) Survival rates of spores in RAW 264.7 cells were assessed using the nystatin
protection assay. (C) The phagocytosis index represents the number of phagocytosed spores per
100 RAW 264.7 cells. (E) LDH releases from infected cells, expressed as a proportion of the high-
control group after 24 h infection. (F) TNF- α secretion by RAW 264.7 cells after infection with
WT and $\Delta ron1$ spores for 10 h. All data are presented as means \pm SD from independent
experimental replicates (ns, no significance; *, $P < 0.05$; **, $P < 0.01$; ***, $P < 0.001$; ****, $P <$
0.0001).

**Discussion**

Opportunistic pathogens, such as *A. fumigatus*, owe their success in part to their exceptional
nutrient acquisition strategies. A key adaptive mechanism is their ability to utilize diverse carbon
sources, which supports survival in hostile environments (52). Despite this, the role of metabolic
pathways in mediating *A. fumigatus*-host interactions remains poorly understood. During
infection, *N*-acetylglucosamine (GlcNAc) is released from the turnover of host macromolecules
like glycosaminoglycans, glycoproteins, and proteoglycans (52). Beyond its role in supporting
growth, GlcNAc is a known virulence factor in various pathogens, including *C albicans*, *C.*
*deneoformans*, and *Leishmania major* (42, 53).

Using the *C. albicans* GlcNAc catabolic gene cluster as a reference, we identified and
characterized three GlcNAc-related genes in *A. fumigatus*: *dac1*, *nag1*, and *ron1*. Deletion of *dac1*
and *nag1* abolished germination and growth on GlcNAc, reconfirming their roles as specialized
enzymes for GlcNAc catabolism (Figure 2 and 3D) (20, 24, 54). However, the $\Delta ron1$ mutant
retained the ability to grow on GlcNAc (Figures 2 and 3D), though with severely impaired growth,
inconsistent with the observations in *C. albicans*, *C tropicalis* and *T. reesei* (17, 20, 24). This
suggests that GlcNAc metabolism in *A. fumigatus* is regulated by mechanisms beyond *ron1*.
Interestingly, osmotic stress and caspofungin exposure restored $\Delta ron1$ mutant growth to wild-type
levels (Figure 2 and Supplementary Figure 2), implicating potential crosstalk between the HOG
(high osmolarity glycerol) (55) and CWI (cell wall integrity) pathways (56) and GlcNAc
utilization, which warrants further investigation.

The $\Delta dac1$ and $\Delta nag1$ mutants exhibited no significant virulence differences in our *G.*
*mellonella* infection model, contrasting with observations in *C. albicans* and *M. oryzae* (23, 42).
This discrepancy may reflect differences in carbon source utilization during infection. While
GlcNAc serves as a primary carbon source for other pathogens (13, 57), *A. fumigatus* appears to
adapt flexibly, utilizing a broad range of carbon sources present in the host environment. As a
result, the defective phenotypes of $\Delta dac1$ and $\Delta nag1$ mutants are restricted to conditions where
GlcNAc is the sole carbon source, without impacting their pathogenicity in more complex
environments (Figure 2, Figure 4 and Supplementary Figure 2).

The $\Delta ron1$ mutant, despite retaining the ability to grow on non-amino sugar carbon sources,

exhibited significantly attenuated virulence. This was further evidenced by reduced survival in
immune cells, increased TNF- α secretion, and upregulation of inflammatory cytokines and
chemokines (Figure 6 and Supplementary Figure 3). These findings suggest that *ron1* influences
*A. fumigatus* virulence through structural changes in conidia. Notably, surface-exposed proteins in
$\Delta ron1$ mutant spores increased by 188% (Figure 5C). WT conidia are coated with RodA, which
confers hydrophobicity and shields the fungus from immune recognition (58, 59). Exposure of cell
wall surface pathogen-associated molecular patterns (PAMPs) triggers quicker immune responses
(31, 49, 60). Similarly, glycoproteins from melanin-deficient mutants ($\Delta pksP$, $\Delta ayg1$, $\Delta arp2$) have
been shown to enhance dendritic cell maturation and cytokine induction (49). The increased
exposing amorphous glycoprotein matrix in $\Delta ags1/\Delta ags2/\Delta ags2$ led to immediate immunity
recognition and TNF- α secretion (31). While the specific proteins exposed in $\Delta ron1$ mutant spores
remain unidentified, their increased immunogenicity likely accounts for the heightened immune
response and reduced virulence (61). Additionally, the reduced melanin content in $\Delta ron1$ mutant
spores may also contribute to their impaired virulence (Figures 5A and 5B). Melanin protects *A.*
*fumigatus* by mitigating phagolysosomal acidification, reactive oxygen species (ROS) killing,
neutrophil recruitment, and chemokine secretion at the post-translational level (35, 62-64). Future
studies will aim to identify the specific proteins exposed on $\Delta ron1$ mutant spores and elucidate the
mechanisms underlying their rapid immune clearance.

In conclusion, this study identifies and characterizes the GlcNAc catabolic pathway in *A.*
*fumigatus*, highlighting *dac1* and *nag1* as key genes for GlcNAc catabolism and *ron1* as a central
regulator. Beyond regulating GlcNAc metabolism, *ron1* also modulates conidial structure by
increasing surface protein exposure and reducing melanin content, leading to faster immune
clearance. These findings establish *ron1* as a potential antifungal target for therapeutic
intervention.

**Author Contributions:** W.F. conceived the study. X.G., X.G., and Q.Q. performed the
experiments. X.G., C.J., and W.F. analyzed and interpreted the data and wrote the manuscript with
input from all authors. All authors have read and agreed on the final manuscript.

**Funding:** This work was supported by Guangxi Natural Science Foundation
(2020GXNSFDA238008, 2023GXNSFFA026011) to W.F. and Basic research fund of Guangxi
Academy of Sciences (2024YWF2111) to X.G., and National Natural Science Foundation of
China (32371339) to W.F.

**Conflict of Interest:** The authors declare no conflict of interest.

**References:**

- 1. Morrissey CO, Kim HY, Duong TN, Moran E, Alastruey-Izquierdo A, Denning DW,
Perfect JR, Nucci M, Chakrabarti A, Rickerts V, Chiller TM, Wahyuningsih R, Hamers
RL, Cassini A, Gigante V, Sati H, Alffenaar JW, Beardsley J. 2024. *Aspergillus*
*fumigatus*-a systematic review to inform the World Health Organization priority list of
fungal pathogens. *Med Mycol* 62.
- 2. Raffa N, Oshero N, Keller NP. 2019. Copper Utilization, Regulation, and Acquisition
by *Aspergillus fumigatus*. *Int J Mol Sci* 20.
- 3. Dagenais TR, Keller NP. 2009. Pathogenesis of *Aspergillus fumigatus* in Invasive
Aspergillosis. *Clin Microbiol Rev* 22:447-65.
- 4. Brown GD, Denning DW, Gow NA, Levitz SM, Netea MG, White TC. 2012. Hidden
killers: human fungal infections. *Sci Transl Med* 4:165rv13.
- 5. Denning DW. 2024. Global incidence and mortality of severe fungal disease. *Lancet*
*Infect Dis* doi:10.1016/s1473-3099(23)00692-8.
- 6. Stevens DA, White TC, Perlin DS, Selitrennikoff CP. 2005. Studies of the paradoxical
effect of caspofungin at high drug concentrations. *Diagn Microbiol Infect Dis* 51:173-8.
- 7. Chowdhary A, Sharma C, Kathuria S, Hagen F, Meis JF. 2015. Prevalence and
mechanism of triazole resistance in *Aspergillus fumigatus* in a referral chest hospital
in Delhi, India and an update of the situation in Asia. *Front Microbiol* 6:428.
- 8. Robbins N, Wright GD, Cowen LE. 2016. Antifungal Drugs: The Current
Armamentarium and Development of New Agents. *Microbiol Spectr* 4.
- 9. Konopka JB. 2012. N-acetylglucosamine (GlcNAc) functions in cell signaling.

- Scientifica (Cairo) 2012.
- 10. Naseem S, Konopka JB. 2015. N-acetylglucosamine Regulates Virulence Properties
in Microbial Pathogens. PLoS Pathog 11:e1004947.
- 11. Barnhart MM, Lynem J, Chapman MR. 2006. GlcNAc-6P levels modulate the
expression of Curli fibers by Escherichia coli. J Bacteriol 188:5212-9.
- 12. Camacho E, Chrissian C, Cordero RJB, Liporagi-Lopes L, Stark RE, Casadevall A.
2017. N-acetylglucosamine affects Cryptococcus neoformans cell-wall composition
and melanin architecture. Microbiology (Reading) 163:1540-1556.
- 13. Yang D, Zhang M, Su C, Dong B, Lu Y. 2023. Candida albicans exploits N-
acetylglucosamine as a gut signal to establish the balance between commensalism
and pathogenesis. Nat Commun 14:3796.
- 14. Pérez-Campo FM, Domínguez A. 2001. Factors affecting the morphogenetic switch in
Yarrowia lipolytica. Curr Microbiol 43:429-33.
- 15. Ye L, Wang S, Zheng J, Chen L, Shen L, Kuang Y, Wang Y, Peng Y, Hu C, Wang L,
Tian X, Liao G. 2022. Functional Characterization of the GlcNAc Catabolic Pathway
in Cryptococcus deneoformans. Appl Environ Microbiol 88:e0043722.
- 16. Du H, Ennis CL, Hernday AD, Nobile CJ, Huang G. 2020. N-Acetylglucosamine
(GlcNAc) Sensing, Utilization, and Functions in Candida albicans. J Fungi (Basel) 6.
- 17. Song YD, Hsu CC, Lew SQ, Lin CH. 2021. Candida tropicalis RON1 is required for
hyphal formation, biofilm development, and virulence but is dispensable for N-
acetylglucosamine catabolism. Med Mycol 59:379-391.
- 18. Shang S, He D, Liu C, Bao X, Han S, Wang L. 2024. TRAF3 gene regulates

- macrophage migration and activation by lung epithelial cells infected with *Aspergillus*
*fumigatus*. *Microbiol Spectr* 12:e0269923.
- 19. Gilmore SA, Naseem S, Konopka JB, Sil A. 2013. N-acetylglucosamine (GlcNAc)
triggers a rapid, temperature-responsive morphogenetic program in thermally
dimorphic fungi. *PLoS Genet* 9:e1003799.
- 20. Kappel L, Gaderer R, Flipphi M, Seidl-Seiboth V. 2016. The N-acetylglucosamine
catabolic gene cluster in *Trichoderma reesei* is controlled by the Ndt80-like
transcription factor RON1. *Mol Microbiol* 99:640-57.
- 21. Kumar MJ, Jamaluddin MS, Natarajan K, Kaur D, Datta A. 2000. The inducible N-
acetylglucosamine catabolic pathway gene cluster in *Candida albicans*: discrete N-
acetylglucosamine-inducible factors interact at the promoter of NAG1. *Proc Natl Acad*
*Sci U S A* 97:14218-23.
- 22. Naseem S, Gunasekera A, Araya E, Konopka JB. 2011. N-acetylglucosamine
(GlcNAc) induction of hyphal morphogenesis and transcriptional responses in
*Candida albicans* are not dependent on its metabolism. *J Biol Chem* 286:28671-
28680.
- 23. Kumar A, Ghosh S, Bhatt DN, Narula A, Datta A. 2016. *Magnaporthe oryzae*
aminosugar metabolism is essential for successful host colonization. *Environ*
*Microbiol* 18:1063-77.
- 24. Naseem S, Min K, Spitzer D, Gardin J, Konopka JB. 2017. Regulation of Hyphal
Growth and N-Acetylglucosamine Catabolism by Two Transcription Factors in
*Candida albicans*. *Genetics* 206:299-314.

- 25. He R, Wei P, Odiba AS, Gao L, Usman S, Gong X, Wang B, Wang L, Jin C, Lu G,
Fang W. 2024. Amino sugars influence *Aspergillus fumigatus* cell wall polysaccharide
biosynthesis, and biofilm formation through interfering galactosaminogalactan
deacetylation. *Carbohydr Polym* 324:121511.
- 26. Du W, Zhai P, Wang T, Bromley MJ, Zhang Y, Lu L. 2021. The C(2)H(2)
Transcription Factor SlrA Contributes to Azole Resistance by Coregulating the
Expression of the Drug Target Erg11A and the Drug Efflux Pump Mdr1 in *Aspergillus*
*fumigatus*. *Antimicrob Agents Chemother* 65.
- 27. Gong X, Zhou Y, Qin Q, Wang B, Wang L, Jin C, Fang W. 2024. Nitrate assimilation
compensates for cell wall biosynthesis in the absence of *Aspergillus fumigatus*
phosphoglucose isomerase. *Appl Environ Microbiol* 90:e0113824.
- 28. Zhou Y, Du C, Odiba AS, He R, Ahamefule CS, Wang B, Jin C, Fang W. 2021.
Phosphoglucose Isomerase Plays a Key Role in Sugar Homeostasis, Stress
Response, and Pathogenicity in *Aspergillus flavus*. *Front Cell Infect Microbiol*
11:777266.
- 29. Fang W, Yu X, Wang B, Zhou H, Ouyang H, Ming J, Jin C. 2009. Characterization of
the *Aspergillus fumigatus* phosphomannose isomerase Pmi1 and its impact on cell
wall synthesis and morphogenesis. *Microbiology (Reading)* 155:3281-3293.
- 30. Zhou Y, Yan K, Qin Q, Raimi OG, Du C, Wang B, Ahamefule CS, Kowalski B, Jin C,
van Aalten DMF, Fang W. 2022. Phosphoglucose Isomerase Is Important for
*Aspergillus fumigatus* Cell Wall Biogenesis. *mBio* 13:e0142622.
- 31. Beauvais A, Bozza S, Kniemeyer O, Formosa C, Balloy V, Henry C, Roberson RW,

Dague E, Chignard M, Brakhage AA, Romani L, Latgé JP. 2013. Deletion of the α -
(1,3)-glucan synthase genes induces a restructuring of the conidial cell wall
responsible for the avirulence of *Aspergillus fumigatus*. PLoS Pathog 9:e1003716.

32. Hu X, Zhou Y, Liu R, Wang J, Guo L, Huang X, Li J, Yan Y, Liu F, Li X, Tan X, Luo Y,
Wang P, Zhou S. 2024. Protein disulfide isomerase 1 is required for RodA
assembling-based conidial hydrophobicity of *Aspergillus fumigatus*. Appl Environ
Microbiol 90:e0126023.

33. Youngchim S, Morris-Jones R, Hay RJ, Hamilton AJ. 2004. Production of melanin by
*Aspergillus fumigatus*. J Med Microbiol 53:175-181.

34. Maubon D, Park S, Tanguy M, Huerre M, Schmitt C, Prévost MC, Perlin DS, Latgé JP,
Beauvais A. 2006. AGS3, an alpha(1-3)glucan synthase gene family member of
*Aspergillus fumigatus*, modulates mycelium growth in the lung of experimentally
infected mice. Fungal Genet Biol 43:366-75.

35. Reedy JL, Jensen KN, Crossen AJ, Basham KJ, Ward RA, Reardon CM, Brown
Harding H, Hepworth OW, Simaku P, Kwaku GN, Tone K, Willment JA, Reid DM,
Stappers MHT, Brown GD, Rajagopal J, Vyas JM. 2024. Fungal melanin suppresses
airway epithelial chemokine secretion through blockade of calcium fluxing. Nat
Commun 15:5817.

36. Ouyang H, Zhang Y, Zhou H, Ma Y, Li R, Yang J, Wang X, Jin C. 2021. Deficiency of
GPI Glycan Modification by Ethanolamine Phosphate Results in Increased Adhesion
and Immune Resistance of *Aspergillus fumigatus*. Front Cell Infect Microbiol
11:780959.

- 37. DuBois M, Gilles KA, Hamilton JK, Rebers PA, Smith F. 1956. Colorimetric Method
for Determination of Sugars and Related Substances. *Analytical Chemistry* 28:350-
356.
- 38. Lee JI, Yu YM, Rho YM, Park BC, Choi JH, Park HM, Maeng PJ. 2005. Differential
expression of the *chsE* gene encoding a chitin synthase of *Aspergillus nidulans* in
response to developmental status and growth conditions. *FEMS Microbiol Lett*
249:121-9.
- 39. Champion OL, Wagley S, Titball RW. 2016. *Galleria mellonella* as a model host for
microbiological and toxin research. *Virulence* 7:840-5.
- 40. Zhang X, Jia X, Tian S, Zhang C, Lu Z, Chen Y, Chen F, Li Z, Su X, Han X, Sun Y,
Han L. 2018. Role of the small GTPase Rho1 in cell wall integrity, stress response,
and pathogenesis of *Aspergillus fumigatus*. *Fungal Genet Biol* 120:30-41.
- 41. Ansari S, Kumar V, Bhatt DN, Irfan M, Datta A. 2022. N-Acetylglucosamine Sensing
and Metabolic Engineering for Attenuating Human and Plant Pathogens.
*Bioengineering (Basel)* 9.
- 42. Singh P, Ghosh S, Datta A. 2001. Attenuation of virulence and changes in
morphology in *Candida albicans* by disruption of the N-acetylglucosamine catabolic
pathway. *Infect Immun* 69:7898-903.
- 43. Yamada-Okabe T, Sakamori Y, Mio T, Yamada-Okabe H. 2001. Identification and
characterization of the genes for N-acetylglucosamine kinase and N-
acetylglucosamine-phosphate deacetylase in the pathogenic fungus *Candida*
*albicans*. *Eur J Biochem* 268:2498-505.

- 44. Su C, Lu Y, Liu H. 2016. N-acetylglucosamine sensing by a GCN5-related N-
acetyltransferase induces transcription via chromatin histone acetylation in fungi. *Nat*
*Commun* 7:12916.
- 45. Vincent F, Davies GJ, Brannigan JA. 2005. Structure and kinetics of a monomeric
glucosamine 6-phosphate deaminase: missing link of the NagB superfamily? *J Biol*
*Chem* 280:19649-55.
- 46. Latgé JP, Chamilos G. 2019. *Aspergillus fumigatus* and Aspergillosis in 2019. *Clin*
*Microbiol Rev* 33.
- 47. Veiga-da-Cunha M, Van Schaftingen E. 2002. Identification of fructose 6-phosphate-
and fructose 1-phosphate-binding residues in the regulatory protein of glucokinase. *J*
*Biol Chem* 277:8466-73.
- 48. Keller S, Macheleidt J, Scherlach K, Schmalzer-Ripcke J, Jacobsen ID, Heinekamp T,
Brakhage AA. 2011. Pyomelanin formation in *Aspergillus fumigatus* requires HmgX
and the transcriptional activator HmgR but is dispensable for virulence. *PLoS One*
6:e26604.
- 49. Bayry J, Beaussart A, Dufrêne YF, Sharma M, Bansal K, Kniemeyer O, Amanianda V,
Brakhage AA, Kaveri SV, Kwon-Chung KJ, Latgé JP, Beauvais A. 2014. Surface
structure characterization of *Aspergillus fumigatus* conidia mutated in the melanin
synthesis pathway and their human cellular immune response. *Infect Immun* 82:3141-
53.
- 50. Li X, Gao M, Han X, Tao S, Zheng D, Cheng Y, Yu R, Han G, Schmidt M, Han L.
2012. Disruption of the phospholipase D gene attenuates the virulence of *Aspergillus*

fumigatus. Infect Immun 80:429-40.

51. Dou YH, Du JK, Liu HL, Shong XD. 2013. The role of procalcitonin in the identification
of invasive fungal infection-a systemic review and meta-analysis. Diagn Microbiol
Infect Dis 76:464-9.

52. Ansari S, Bhatt DN, Sood C, Datta A. 2021. Functional characterization of the
LdNAGD gene in Leishmania donovani. Microbiol Res 251:126830.

53. Naderer T, Heng J, McConville MJ. 2010. Evidence that intracellular stages of
Leishmania major utilize amino sugars as a major carbon source. PLoS Pathog
6:e1001245.

54. Bhatt DN, Ansari S, Kumar A, Ghosh S, Narula A, Datta A. 2020. Magnaporthe
oryzae MoNdt80 is a transcriptional regulator of GlcNAc catabolic pathway involved in
pathogenesis. Microbiol Res 239:126550.

55. Yaakoub H, Sanchez NS, Ongay-Larios L, Courdavault V, Calenda A, Bouchara JP,
Coria R, Papon N. 2022. The high osmolarity glycerol (HOG) pathway in fungi(†). Crit
Rev Microbiol 48:657-695.

56. Dichtl K, Samantaray S, Wagener J. 2016. Cell wall integrity signalling in human
pathogenic fungi. Cell Microbiol 18:1228-38.

57. Min K, Naseem S, Konopka JB. 2019. N-Acetylglucosamine Regulates
Morphogenesis and Virulence Pathways in Fungi. J Fungi (Basel) 6.

58. Aimanianda V, Bayry J, Bozza S, Kniemeyer O, Perruccio K, Elluru SR, Clavaud C,
Paris S, Brakhage AA, Kaveri SV, Romani L, Latgé JP. 2009. Surface hydrophobin
prevents immune recognition of airborne fungal spores. Nature 460:1117-21.

- 59. Bayry J, Aimanianda V, Guijarro JI, Sunde M, Latgé JP. 2012. Hydrophobins--unique
fungal proteins. PLoS Pathog 8:e1002700.
- 60. Alsteens D, Aimanianda V, Hegde P, Pire S, Beau R, Bayry J, Latgé JP, Dufrêne YF.
2013. Unraveling the nanoscale surface properties of chitin synthase mutants of
*Aspergillus fumigatus* and their biological implications. Biophys J 105:320-7.
- 61. Bernard M, Mouyna I, Dubreucq G, Debeaupuis JP, Fontaine T, Vorgias C, Fuglsang
C, Latgé JP. 2002. Characterization of a cell-wall acid phosphatase (PhoAp) in
*Aspergillus fumigatus*. Microbiology (Reading) 148:2819-2829.
- 62. Langfelder K, Jahn B, Gehringer H, Schmidt A, Wanner G, Brakhage AA. 1998.
Identification of a polyketide synthase gene (pksP) of *Aspergillus fumigatus* involved
in conidial pigment biosynthesis and virulence. Med Microbiol Immunol 187:79-89.
- 63. Amin S, Thywissen A, Heinekamp T, Saluz HP, Brakhage AA. 2014. Melanin
dependent survival of *Aspergillus fumigatus* conidia in lung epithelial cells. Int J Med
Microbiol 304:626-36.
- 64. Tsai HF, Chang YC, Washburn RG, Wheeler MH, Kwon-Chung KJ. 1998. The
developmentally regulated alb1 gene of *Aspergillus fumigatus*: its role in modulation
of conidial morphology and virulence. J Bacteriol 180:3031-8.

A**B****C**

A**B****C****D****E**
PERCENT

F
A**B****C****D****E****F**
9th feb 2025

Review paper by Gong et al. (2025)

In the paper by Gong et al. is afforded the study of GlcNAc catabolism and its relevance in virulence of *A. fumigatus* based on previous studied on other fungal pathogens. Although this study focuses on a topic well known in other pathogens, it is worthy to ascertain whether GlcNAc catabolism in *A. fumigatus* functions like in other fungal pathogens or it has some distinctive feature that somewhat enhances the significance of this process in the virulence of *A. fumigatus*. However, this paper has major weaknesses that should be addressed before publication:

- 1) Although mutants were constructed in a KU80 genetic background, **they should be analyzed by southern blot** to preclude ectopic integration of transformant DNA in any other loci. In addition, **reconstituted strains should be generated for each mutant as the best controls for all observed phenotypic traits.**
- 2) Figure 1B does not reflect the actual arrangement of genes *dac1*, *nag1* and *ron1* in the *A. fumigatus* genome and the alphanumeric codes are wrong (also in table 1): AFUB_083510 is *ron1* (not *dac1*) and AFUB_08346 is *dac1* (not *ron1*). Also, the schematic representation of the locus shown in figure 1B suggests erroneously that these genes are in a cluster. Although they are closely located one each other in the genome, they are separated by other genes. Hence, the **figure 1B should be rebuilt** including the all genes between them and including a scale.
- 3) Lines 278-295. Testing the effect of different stressing compounds (except sorbitol and salts) on fungal mutant strains grown in the presence of GlcNAc as sole carbon source does not make sense since these mutants do not grow under these culture conditions. They should be better tested in the presence of glucose as sole carbon source (as shown for antifungals in Fig.S2). In addition, most phenotypic descriptions are not accurate leading to misinterpretations. For example, in line 282-3 it is stated that “The Δ ron1 mutant was more sensitive to CR and CFW, indicating the cell wall integrity was impaired”, but this is not true if considered that wild-type strain also grows worse in the presence of CR and CFW than in their absence. The effect of sorbitol and salts, which could protect fungal strains against osmotic lysis in hypotonic media, argue in favor of a role of Ron1 in maintaining cell wall integrity. However, the statement in line 291 “ron1 is crucial in maintaining cell wall integrity” is an overinterpretation of results. To show that this is as stated, the strains should be grown with sorbitol or salts in the presence of a carbon source other than GlcNAc to tests whether the level of GlcNAc synthesized intracellularly and used for chitin biosynthesis is regulated by Ron1.

Other points to be addressed are the following:

- 1) Line 199. Similarities or identities? Inconsistence with Table 1? It should be indicated both. Indicate also the alphanumeric codes for these genes in the reference fungal strain Af293. In case there were any noticeably difference between A1163 and Af293 regarding these genes, it should be discussed. Genes should not be named as *dac1*, *nag1* and *ron1*. Instead, they should be named as *dacA*, *nagA* and *ronA* throughout the text according to rule most largely accepted for gene nomenclature in filamentous fungi.
- 2) Line 80-81. The notation of the strain CEA17 genotype is badly written (is not Δ ku80 Δ pyrG-). CEA17 is an auxotrophic for Uracil (PyrG-) because it has a substitution (not a deletion) in a nucleotide of the *pyrG* gene and it carries the wild-type *akuB* (KU80) gene. The CEA17 derived Δ akuB and PyrG- fungal strain was created by Silva Ferreira et

- al. (2006, Eukaryotic Cell 5:207-211) and named as KU80 Δ pyrG (reference not cited in the text).
- 3) Line 115. Growth curves for filamentous fungi growing in liquid media cannot be measured by OD (during 5 days!!!).
 - 4) Line 157-188. Indicate the culture media used for all these experiments.
 - 5) Line 180. After washing with PBS?
 - 6) Line 174. What means high control?
 - 7) Line 200. In figure 1B is not show any comparison but the catabolic cluster for GlcNAc.
 - 8) Line 202 and figure 1C. It is not clear at all how expression levels were relativized. What means gene/tbp?
 - 9) Line 221 and from now on. Is GlcN or GlcN6P?
 - 10) Line 222. For me the expression “impaired growth” is the same that “growth was completely inhibited”. Figure 2 should be cited at this point.
 - 11) Line 224. The Δ ron1 mutant does not act as a transcription factor (the transcription factor is Ron1). You should pay more attention in wording properly.
 - 12) Lines 236-273. This is worthless and could be deleted.
 - 13) Line 278. Could this explain the difference in the grow between wild-type and each mutant with glucose, fructose, mannose, sucrose and xylose as sole carbon sources as seen in figure 2?
 - 14) Lines 329-330. Not consistent with data shown in Fig. 6B.

Reviewer #1 (Comments for the Author):

In this work, Gong and coworkers characterize functionally three proteins of *Aspergillus fumigatus* necessary in N-acetylglucosamine metabolism, with special focus on the transcription factor *ron1*. The authors analyzed expression levels of these three genes, the phenotypes in multiple culture media for their null mutants, germination patterns, conidia production, cell-wall and conidial outer layer composition, and virulence.

In my opinion, the work is of interest for the readers of mSpectrum, but it needs improvement.

First, please, check writing (e.g., in the title, replace "drived" by "driven").

Response: Thank you for your nice comments and suggestions. We have modified the title as “Transcription Factor RonA-Driven GlcNAc Catabolism Is Essential for Growth, Cell Wall Integrity, and Pathogenicity in *Aspergillus fumigatus*” in the revised manuscript.

Second, the authors describe that the null mutants generated only show clear phenotypes when grown on GlcNAc or GlcN, but I see growth and conidiation phenotypes on other media, such as when xylose or mannose are used as carbon sources. How do the authors explain that? I would recommend quantification of radial growth and conidia production, not with all inocula tested in Figure 2 but when 1000 or 10,000 conidia are inoculated.

Response: Thank you for your comment. We agree that the null mutants show mild phenotypes on other carbon sources. To address this, we constructed and verified the RT strains and performed spot assays with 10,000 conidia per plate (revised Figure 2). Additional assays on other single carbon sources are shown in Supplementary Figure 2. All strains showed similar growth under these conditions, indicating that the phenotypes are more pronounced under GlcNAc or GlcN conditions.

Third, some of the quantification criteria used by the authors should be more consistent. For example, in lines 155-156, "The internalization rate was calculated as the percentage of conidia colonies relative to initial inoculum." Is this a reliable method for quantification? How do the authors guarantee that these colonies on MM plates developed from single and isolated conidia?.

Response: Thank you for your thoughtful comment. Prior to the experiment, we confirmed by microscopy that conidia were individually resuspended and that nearly all conidia were internalized by host cells within 2 hours of co-incubation. After cell lysis, we also examined the samples microscopically to ensure conidia remained largely separated. The internalization rate was calculated based on the number of resulting colonies relative to the initial inoculum, following methods described in previously published studies (Shang et al., *Microbiol Spectr*, 2024; Zhang et al., *Fungal Genet Biol*, 2018; Jia et al., *Front Microbiol*, 2018; Li et al., *Infect Immun*, 2011). This assay was performed in triplicate and under conditions consistent with those references.

And finally, I saw no complementation or overexpression strain for the genes analyzed in this work. Wouldn't these strains be necessary?

Response: Thank you for your suggestion. We agree that revertant strains are important for validating gene function. In this revised manuscript, we have included the construction and verification of the revertant strains, confirmed by PCR and Southern blot (see Supplementary Figure 1).

I attach to my report a pdf copy of the manuscript including all my comments and suggestions. Hope it will help.

Response: Thank you very much for taking the time to provide detailed feedback and for attaching the annotated PDF. Your comments and suggestions are greatly appreciated and will be very helpful in improving the quality of the manuscript.

Reviewer #2 (Comments for the Author):

In the paper by Gong et al. is afforded the study of GlcNAc catabolism and its relevance in virulence of *A. fumigatus* based on previous studied on other fungal pathogens. Although this study focuses on a topic well known in other pathogens, it is worthy to ascertain whether GlcNAc catabolism in *A. fumigatus* functions like in other fungal pathogens or it has some distinctive feature that somewhat enhances the significance of this process in the virulence of *A. fumigatus*. However, this paper has major weaknesses that should be addressed before publication:

1) Although mutants were constructed in a KU80 genetic background, they should be analyzed by southern blot to preclude ectopic integration of transformant DNA in any other loci. In addition, reconstituted strains should be generated for each mutant as the best controls for all observed phenotypic traits.

Response: Thank you for your valuable suggestions. To ensure correct gene replacement and exclude ectopic integration, all three mutant strains were verified by Southern blot (Supplementary Figures 1E, 1F, and 1G). For each mutant, the corresponding gene and the *pyrG* marker were reintroduced to construct revertant strains, which were verified using five pairs of diagnostic primers and Southern blot (Supplementary Figures 1B, 1C, 1D and 1G).

2) Lines 278-295. Testing the effect of different stressing compounds (except sorbitol and salts) on fungal mutant strains grown in the presence of GlcNAc as sole carbon source does not make sense since these mutants do not grow under these culture conditions. They should be better tested in the presence of glucose as sole carbon source (as shown for antifungals in Fig.S2).

Response: Thank you for your helpful suggestion. Following your advice, we tested the sensitivity of the mutant strains to various stressors using glucose as the sole carbon source. As shown in Figure 3A, only the $\Delta dacA$ mutant exhibited slight sensitivity to CFW and NaCl,

while the other strains showed no significant differences.

In addition, most phenotypic descriptions are not accurate leading to misinterpretations. For example, in line 282-3 it is stated that "The Δ ron1 mutant was more sensitive to CR and CFW, indicating the cell wall integrity was impaired", but this is not true if considered that wild-type strain also grows worse in the presence of CR and CFW than in their absence.

Response: Thank you for your comment. We agree that the wild-type strain also shows reduced growth in the presence of CR and CFW. However, under identical conditions, the Δ ronA mutant exhibited markedly higher sensitivity. Specifically, on CR-containing media, 10^3 wild-type conidia were able to grow, whereas Δ ronA showed no visible growth. Similarly, on CFW, both 10^3 and 10^2 wild-type conidia grew, while Δ ronA failed to grow. These observations suggest that the Δ ronA mutant is indeed more sensitive to CR and CFW compared to the wild type, indicating compromised cell wall integrity.

The effect of sorbitol and salts, which could protect fungal strains against osmotic lysis in hypotonic media, argue in favor of a role of Ron1 in maintaining cell wall integrity. However, the statement in line 291 "ron1 is crucial in maintaining cell wall integrity" is an overinterpretation of results. To show that this is as stated, the strains should be grown with sorbitol or salts in the presence of a carbon source other than GlcNAc to tests whether the level of GlcNAc synthesized intracellularly and used for chitin biosynthesis is regulated by Ron1.

Response: Thank you for pointing this out. We agree that the statement was an overinterpretation. We have revised the text to: "*ronA* is important for maintaining cell wall integrity under GlcNAc conditions," to more accurately reflect the data and experimental context.

3) Figure 1B does not reflect the actual arrangement of genes *dac1*, *nag1* and *ron1* in the *A. fumigatus* genome and the alphanumeric codes are wrong (also in table 1): AFUB_083510 is *ron1* (not *dac1*) and AFUB_08346 is *dac1* (not *ron1*). Also, the schematic representation of the locus shown in figure 1B suggests erroneously that these genes are in a cluster. Although they are closely located one each other in the genome, they are separated by other genes. Hence, the figure 1B should be rebuilt including the all genes between them and including a scale.

Response: Thank you for your valuable suggestion. We have carefully reviewed the gene annotations using NCBI. AFUB_083460 corresponds to *dacA*, AFUB_083490 to *nagA*, and AFUB_083510 to *ronA*. These genes involved in GlcNAc metabolism are not organized in a gene cluster. Accordingly, we have redesigned Figure 1C to accurately represent the genomic arrangement, including all intervening genes and maintaining proportional scaling of gene sizes and intergenic distances based on their actual base-pair lengths.

Other points to be addressed are the following:

1) Line 199. Similarities or identities? Inconsistence with Table 1? It should be indicated both. Indicate also the alphanumeric codes for these genes in the reference fungal strain Af293. In case there were any noticeable difference between A1163 and Af293 regarding these genes, it should be discussed. Genes should not be named as *dac1*, *nag1* and *ron1*. Instead, they should be named as *dacA*, *nagA* and *ronA* throughout the text according to rule most largely accepted for gene nomenclature in filamentous fungi.

Response: Thank you for your helpful suggestion. In the revised manuscript, we have included both similarity and identity values for the GlcNAc catabolic genes in strains A1163 and Af293, along with their corresponding alphanumeric codes. Interestingly, Af293 contains two homologues of these genes, while A1163 has a single copy of each. Additionally, we have updated the gene names throughout the manuscript to follow the widely accepted nomenclature for filamentous fungi, using *dacA*, *nagA* and *ronA* instead of *dac1*, *nag1*, and *ron1*.

2) Line 80-81. The notation of the strain CEA17 genotype is badly written (is not $\Delta ku80\Delta pyrG^-$). CEA17 is an auxotrophic for Uracil ($PyrG^-$) because it has a substitution (not a deletion) in a nucleotide of the *pyrG* gene and it carries the wild-type *akuB* (KU80) gene. The CEA17 derived $\Delta akuB$ and $PyrG^-$ fungal strain was created by Silva Ferreira et al. (2006, Eukaryotic Cell 5:207-211) and named as $KU80\Delta pyrG$ (reference not cited in the text).

Response: Thank you for your valuable suggestion. We have corrected as below for clear description: “The *A. fumigatus* strain CEA17 is an uracil auxotrophic strain carrying a point mutation in the *pyrG* gene, resulting in a *pyrG* phenotype. Importantly, CEA17 retains the wild-type *akuB* (also known as *KU80*) gene. A derivative strain lacking *akuB* ($\Delta akuB$) and also *pyrG* was constructed by Silva Ferreira et al. (2006, Eukaryotic Cell 5:207–211) and designated as $KU80\Delta pyrG$. This strain is commonly used to facilitate targeted gene deletions due to the increased efficiency of homologous recombination resulting from *akuB* deletion”.

3) Line 115. Growth curves for filamentous fungi growing in liquid media cannot be measured by OD (during 5 days!!!).

Response: Thank you for your suggestion. We have removed the growth curve measurement by OD from the revised manuscript, as it is not appropriate for filamentous fungi grown in liquid media over extended periods.

4) Line 157-188. Indicate the culture media used for all these experiments.

Response: Thank you for your suggestion. We have now specified the culture medium used for these experiments in the “Materials and Methods” section of the revised manuscript: complete DMEM supplemented with 10% FBS, 0.1% streptomycin, and 0.1% gentamicin.

5) Line 180. After washing with PBS?

Response: Yes, the plates were washed multiple times with PBS, following the protocols described in the reference (Shang S *et al*; Microbiol Spectr, 2024; Fungal Genet Biol, 2018).

6) Line 174. What means high control?

Response: The term "high control" refers to the positive control defined by the assay kit. It consists of untreated cells that are lysed using the provided lysis buffer to represent maximum signal or response.

7) Line 200. In figure 1B is not show any comparison but the catabolic cluster for GlcNAc.

Response: Thank you for your comment. Figure 1B was intended to show the genomic organization of *nagA*, *dacA*, and *ronA* on chromosome 8, not a comparison with *Candida albicans*. We have revised it as Figure 1C in the revised manuscript.

8) Line 202 and figure 1C. It is not clear at all how expression levels were relativized. What means gene/tbp?

Response: Thank you for your comment. Figure 1C (now is 1B in revised version) shows qRT-PCR results, where the relative expression levels of *nagA*, *dacA* and *ronA* were calculated using the $2^{-\Delta\Delta ct}$ method. The gene *tbp* was used as the internal reference for normalization. We have cited the original method by Livak and Schmittgen (Livak KJ, Schmittgen TD, Methods, 2001) in the revised manuscript.

9) Line 221 and from now on. Is GlcN or GlcN6P?

Response: GlcN was used as the carbon source in the media. Upon uptake, GlcN can be phosphorylated intracellularly to GlcN-6-phosphate (GlcN6P).

10) Line 222. For me the expression "impaired growth" is the same that "growth was completely inhibited". Figure 2 should be cited at this point.

Response: Thank you for your suggestion. We cited Figure 2 here.

11) Line 224. The Δ ron1 mutant does not act as a transcription factor (the transcription factor is Ron1). You should pay more attention in wording properly.

Response: Thank you for your careful reading and helpful comment. We clarify that RonA is the transcription factor, not the Δ ron1 mutant. We have revised the wording accordingly in the updated manuscript to avoid any confusion.

12) Lines 236-273. This is worthless and could be deleted.

Response: Thank you for your suggestion. We have removed this part.

13) Line 278. Could this explain the difference in the grow between wild-type and each mutant with glucose, fructose, mannose, sucrose and xylose as sole carbon sources as seen in figure 2?

Response: Thank you for your thoughtful question. We also observed that the Δ *dacA*, Δ *nagA* and Δ *ronA* mutant as well as their respective reconstituted strains, exhibited slight growth differences on certain carbon sources. Interestingly, the reconstituted strains did not fully

restore wild-type growth. This may be due to differences in gene expression regulation or protein expression profiles between the reconstituted and wild-type strains. Further investigation is needed to clarify this observation.

14) Lines 329-330. Not consistent with data shown in Fig. 6B.

Response: Thank you for pointing this out. The figure citation is correct. As shown in Figure 6B, the internalization rate of the WT strain is $18.6\% \pm 3.7$, while that of the $\Delta ronA$ mutant is $2.3\% \pm 0.9$. This represents an 87.6% reduction in internalization relative to the WT, calculated as $(18.6\% - 2.3\%) / 18.6\%$.

Re: Spectrum00122-25R1 (Transcription Factor RON1-Driven GlcNAc Catabolism Is Essential for Growth, Cell Wall Integrity, and Pathogenicity in *Aspergillus fumigatus*)

Dear Prof. Wenxia FANG:

Thank you for the privilege of reviewing your work.

I have now received reports from two colleagues on your revision of your former submission. One of the referees recommends acceptance. However, the second referee calls my attention to the fact that many of his/her concerns have not been addressed in this revised version, which in his/her opinion would justify rejection. Those concerns are listed in the accompanying report. Given the conflicting opinions, I will give you the opportunity of submitting another revised version, in the understanding that you should address in a satisfactory manner each and every concern raised by the critic referee.

Revision Guidelines

Sincerely,
Miguel Penalva
Editor
Microbiology Spectrum

Reviewer #1 (Comments for the Author):

The authors have replied to all my queries and have addressed them.

Reviewer #2 (Comments for the Author):

It is well appreciate that the authors carried out some of the requested work, namely construction of revertant strains to be used as controls, but I feel that the authors have lost the chance to improve the manuscript. Actually, redesigned figures, wording, use of English and global style of the revised version is very far from being adequate for publication. These are a few examples:

1. Most responses and explanations to this reviewer were not included later in the revised version of the text.
2. In supplementary figure 1 the scheme to explain construction of mutant and revertant strains is good for a general explanation of the procedure, but not for the specific strains created for this study. Besides the nomenclature of oligonucleotides is very confusing, and were indicated neither the southern blot procedure nor probes used for southern blot and restriction enzymes to digest gDNA for southern blot analyses. It is not possible to follow strain construction.
3. It is intriguing that the size of the colonies for all mutant and revertant strains are smaller than the wild-type colonies in MM+Glc (Fig.2).
4. Figure 3 is the same as supplemental figure 3.
5. Revertant strains were not included as controls in Fig.3 and in experiments where a statistically significant difference between a mutant and wild-type were observed (e.g. Fig.4A,B,C,D; Fig.5A,B,C,E,F and Fig.S4).
6. In the review version has been stated "The *A. fumigatus* strain CEA17 is an uracil auxotrophic strain carrying a point mutation in the *pyrG* gene, resulting in a *pyrG*- phenotype". This statement was not quoted properly (d'Enfert, 1996). In addition, in lines 87 and 101 you insist on mentioning the use of the CEA17 strain when what you actually used was the Δ akuB *PyrG*- fungal strain constructed by Silva Ferreira et al. (2006).
7. In RT-qPCR is not indicated what is *tbp* as required. Is this the TATA binding protein encoding gene? If so, why is this a good internal reference for *Aspergillus fumigatus*? A reference to support this assumption and gene code for *tbp* would be required.
8. Composition of minimal medium must be provided since there are many different formulations for this medium.
9. ORFs in figure 1C (1B as indicated in the text of the revised version) should include both gene codes and gene name (if any).
10. It was not discussed that the three genes are duplicated in Af293.
11. Etc...

Reviewer #2

2nd Review of paper by Gong et al. (2025)

It is well appreciate that the authors carried out some of the requested work, namely construction of revertant strains to be used as controls, but I feel that the authors have lost the chance to improve the manuscript. Actually, redesigned figures, wording, use of English and global style of the revised version is very far from being adequate for publication. These are a few examples:

1. Most responses and explanations to this reviewer were not included later in the revised version of the text.
2. In supplementary figure 1 the scheme to explain construction of mutant and revertant strains is good for a general explanation of the procedure, but not for the specific strains created for this study. Besides the nomenclature of oligonucleotides is very confusing, and were indicated neither the southern blot procedure nor probes used for southern blot and restriction enzymes to digest gDNA for southern blot analyses. It is not possible to follow strain construction.
3. It is intriguing that the size of the colonies for all mutant and revertant strains are smaller than the wild-type colonies in MM+Glc (Fig.2).
4. Figure 3 is the same as supplemental figure 3.
5. Revertant strains were not included as controls in Fig.3 and in experiments where a statistically significant difference between a mutant and wild-type were observed (e.g. Fig.4A,B,C,D; Fig.5A,B,C,E,F and Fig.S4).
6. In the review version has been stated that “The *A. fumigatus* strain CEA17 is an uracil auxotrophic strain carrying a point mutation in the *pyrG* gene, resulting in a *pyrG*⁻ phenotype”. This statement was not quoted properly (d’Enfert, 1996). In addition, in lines 87 and 101 you insist on mentioning the use of the CEA17 strain when what you actually used was the \DeltaakuB *PyrG*⁻ fungal strain constructed by Silva Ferreira et al. (2006).
7. In RT-qPCR is not indicated what is *tbp* as required. Is this the TATA binding protein encoding gene? If so, why is this a good internal reference for *Aspergillus fumigatus*? A reference to support this assumption and gene code for *tbp* would be required.
8. Composition of minimal medium must be provided since there are many different formulations for this medium.
9. ORFs in figure 1C (1B as indicated in the text of the revised version) should include both gene codes and gene name (if any).
10. It was not discussed that the three genes are duplicated in Af293.
11. Etc...

Reviewer #2

2nd Review of paper by Gong et al. (2025)

It is well appreciate that the authors carried out some of the requested work, namely construction of revertant strains to be used as controls, but I feel that the authors have lost the chance to improve the manuscript. Actually, redesigned figures, wording, use of English and global style of the revised version is very far from being adequate for publication. These are a few examples:

1. Most responses and explanations to this reviewer were not included later in the revised version of the text.

Response: Thank you for your comment. We carefully addressed each of the reviewer's questions in the revised manuscript. If any specific points remain unclear or were inadvertently omitted, we would appreciate it if you could indicate them, and we will be happy to revise accordingly.

2. In supplementary figure 1 the scheme to explain construction of mutant and revertant strains is good for a general explanation of the procedure, but not for the specific strains created for this study. Besides the nomenclature of oligonucleotides is very confusing, and were indicated neither the southern blot procedure nor probes used for southern blot and restriction enzymes to digest gDNA for southern blot analyses. It is not possible to follow strain construction.

Response: Thank you for your valuable feedback. To improve clarity and traceability, the revised Supplementary Figure 1 now presents separate schematic diagrams detailing the genetic modifications for each strain: $\Delta dacA$ (1A), $\Delta nagA$ (1D), and $\Delta ronA$ (1G). We have also revised the primer nomenclature to assign a unique, consistent name to each oligonucleotide, and their functions and applications (e.g., for strain verification or Southern blot analysis) are now clearly annotated in Supplementary Table 1. Additionally, detailed descriptions of the Southern blot procedures—including the restriction enzymes used and the design of hybridization probes—have been added to the Materials and Methods section of the revised manuscript.

3. It is intriguing that the size of the colonies for all mutant and revertant strains are smaller

than the wild-type colonies in MM+Glc (Fig.2).

Response: Thank you for your comments. We also noticed that the RT strains did not fully restore the wild-type phenotypes. However, both PCR and Southern blot analyses confirmed the correct construction of these strains, with no evidence of ectopic insertion.

Incomplete phenotypic restoration by revertant strains has been reported in *A. fumigatus* gene function studies. For example, partial restoration has been documented in studies of:

- Growth of WT, *sebA*, and *sebA::sebA⁺* strains on MM and MM plus 10% FBS (Dinamarco TM et al., Eukaryot Cell, 2012)
- Germination rates (Ting Du et al., Fungal Genet Biol, 2019)
- Conidiation (Guoxing Zhu et al., Microbiol Res, 2024)
- Growth and virulence phenotypes (Winkelströter LK et al., Mol Microbiol, 2015)
- Virulence and immune response assays in *nsdC::nsdC⁺* strains (Alves de Castro P et al., mBio, 2021)

This partial complementation may result from metabolic changes or regulatory imbalances that persist even after gene reintroduction. We have now clarified this point in the revised manuscript.

4. Figure 3 is the same as supplemental figure 3.

Response: Thank you for pointing this out, and we apologize for the oversight. We have corrected this error in the revised manuscript by replacing the duplicated figure.

5. Revertant strains were not included as controls in Fig.3 and in experiments where a statistically significant difference between a mutant and wild-type were observed (e.g. Fig.4A,B,C,D; Fig.5A,B,C,E,F and Fig.S4).

Response: We appreciate your thoughtful comment. In principle, revertant (RT) strains should indeed be included as controls for phenotypic validation. In our study, the *ronA* deletion and revertant strains (rRT) were both rigorously verified through PCR and Southern blot analyses (see Supplementary Figure 1). In Figure 2, the rRT strain was shown to restore conidial pigmentation similar to the wild type, indicating successful complementation of the cell wall structure.

The phenotypic differences observed in Figures 4, 5, and Supplementary Figure S4 are

primarily attributed to alterations in the conidial cell wall resulting from the *ronA* deletion. Since the rRT strain largely restored these cell wall-associated features, we focused our comparisons on the wild-type and mutant strains in subsequent functional assays.

6. In the review version has been stated that “The *A. fumigatus* strain CEA17 is an uracil auxotrophic strain carrying a point mutation in the *pyrG* gene, resulting in a *pyrG*⁻ phenotype”. This statement was not quoted properly (d’Enfert, 1996). In addition, in lines 87 and 101 you insist on mentioning the use of the CEA17 strain when what you actually used was the DakuB *Pyrg*⁻ fungal strain constructed by Silva Ferreira et al. (2006).

Response: Thank you for pointing this out. We have corrected the citation of d’Enfert (1996) in the revised manuscript to ensure proper attribution. Additionally, you are correct that the $\DeltaakuB^{KU80}/pyrG^-$ strain described by Silva Ferreira et al. (2006), and not CEA17, was used for protoplast preparation and transformation. This has been clarified and corrected in the revised version of the manuscript.

7. In RT-qPCR is not indicated what is *tbp* as required. Is this the TATA binding protein encoding gene? If so, why is this a good internal reference for *Aspergillus fumigatus*? A reference to support this assumption and gene code for *tbp* would be required.

Response: Thank you for your comment. TBP refers to the TATA-binding protein, which is commonly used as a reference (housekeeping) gene in RT-qPCR studies due to its relatively stable expression. It is one of the seven widely accepted housekeeping genes (including 18S, 28S, ACT, GAPDH, EF1 α , RPL7, and TBP) used for normalization in gene expression analyses (Meller et al., Placenta, 2005). TBP has also been validated and used as an internal reference gene in *Aspergillus fumigatus* and other fungal species in several published studies (e.g., Zhang YW et al., Appl Environ Microbiol, 2020; Langnaese K et al., BMC Mol Biol, 2008; Liu Q et al., Molecules, 2018). These references and the corresponding gene code for *A. fumigatus* TBP have been added to the revised manuscript for clarification.

8. Composition of minimal medium must be provided since there are many different formulations for this medium.

Response: Thank you for your comment. We have now provided a detailed composition of the minimal medium used in our experiments in the revised manuscript to ensure clarity and reproducibility.

9. ORFs in figure 1C (1B as indicated in the text of the revised version) should include both

gene codes and gene name (if any).

Response: Thank you for your comment. Following your suggestion, we have updated Figure 1C to include both the gene codes and their corresponding gene names (where available). Specifically, *AFUB_083470* encodes a GMC family oxidoreductase, *AFUB_083480* encodes betaine aldehyde dehydrogenase, and *AFUB_083500* encodes a putative beta-N-acetylglucosaminidase. These annotations are now clearly indicated in Figure 1C and described in the figure legend.

10. It was not discussed that the three genes are duplicated in Af293.

Response: Thank you for your comment. We have added the following discussion to the manuscript:

Af293 is a clinical isolate obtained in 1993 from a lung biopsy (Nierman WC et al., *Nature*, 2005), while A1163 is a derivative of the clinical isolate CEA10 (d'Enfert C et al., *Curr Genet*, 1996). Phylogenetic analysis of 169 *A. fumigatus* genomes classified Af293 in cluster III and A1163 in cluster I (Garcia-Rubio R et al., *Genes*, 2018). Although both strains have eight chromosomes and over 97% gene identity, significant genetic variation exists (Fedorova ND et al., *PLoS Genet*, 2008). This genetic diversity explains why Af293 harbors two copies of the GlcNAc metabolic genes, whereas A1163 contains only one set. Further functional studies are planned to determine which gene copies in Af293 are responsible for GlcNAc catabolism.

Re: Spectrum00122-25R2 (Transcription Factor RON1-Driven GlcNAc Catabolism Is Essential for Growth, Cell Wall Integrity, and Pathogenicity in *Aspergillus fumigatus*)

Dear Prof. Wenxia FANG:

Thank you very much for submitting this revised version. One referee found that there are several pending minor issues that will not be very difficult to address. Among them is the inclusion of appropriate controls in the indicated figures.

Revision Guidelines

Sincerely,
Miguel Penalva
Editor
Microbiology Spectrum

Reviewer #2 (Comments for the Author):

Most questions have been responded properly by the authors and the manuscript has gained clarity. However, some concerns still remains unaddressed:

Major concern:

The revertant strain for the Δ ronA mutant must be necessarily included at least in Figs.4B, 4C and 5A to validate the results.

Minor questions:

- a) Line 123. What is 5-Fluprppratoc?
- b) Line 142-143. Must be indicated the downstream coding regions detected by the different probes (also in the pictures)
- c) Line 154. Indicate concentration of the different sugars in culture media.
- d) Line 249. Is it Figure B instead of figure C?
- e) Line 250-256. This text must be included in the discussion section.
- f) Lines 317-321. Panels of figure 4 should be labeled accordingly.
- g) Line 379. Is supplemental figure 5 or 4?

Reviewer #2 (Comments for the Author):

Most questions have been responded properly by the authors and the manuscript has gained clarity. However, some concerns still remains unaddressed:

Major concern:

The revertant strain for the $\Delta ronA$ mutant must be necessarily included at least in Figs.4B, 4C and 5A to validate the results.

Thank you for your insightful comment. As suggested, we have included the revertant (rRT) strain in the melanin extraction, spore surface protein comparison, and virulence assays alongside the WT and $\Delta ronA$ strains. The results demonstrated that the rRT strain phenocopied the WT, thereby confirming that the observed phenotypes were specific to the $\Delta ronA$ mutant. These new data are now incorporated into the revised Figures 4A–C and 5A.

Minor questions:

a) Line 123. What is 5-Fluprpratoc?

Response: Thank you for pointing this out. The correct term is 5-fluoroorotic acid (5-FOA). The *pyrG* gene encodes orotidine-5'-phosphate decarboxylase (OMP decarboxylase), an essential enzyme in the de novo pyrimidine biosynthesis pathway. While 5-FOA itself is not toxic, it is converted into the toxic compound 5-fluorouracil (5-FU) through enzymatic reactions that require a functional *pyrG* gene. Therefore, only *pyrG*-deficient strains can grow on media supplemented with 5-FOA together with uracil and uridine.

b) Line 142-143. Must be indicated the downstream coding regions detected by the different probes (also in the pictures).

Response: Thank you for your valuable feedback. We have now indicated the Southern blot probes in the revised Figures 1A, 1D, and 1G. Specifically, the ~1 kb probes for verifying the $\Delta dacA$ and $\Delta nagA$ deletions were the genomic regions immediately downstream of the respective genes. For the $\Delta ronA$ deletion, the probe begins ~100 bp downstream of the gene's stop codon and spans approximately 1 kb.

c) Line 154. Indicate concentration of the different sugars in culture media.

Response: Thank you for pointing this out. All carbon sources added to the medium were used at a final concentration of 1% (w/v). This information has been revised and included in the submitted manuscript.

d) Line 249. Is it Figure B instead of figure C?

Response: Thank you for noticing this mistake. We renamed Figure 1 accordingly. The correct reference is Figure 1C, not Figure 1B, and this has been corrected in the revised manuscript.

e) Line 250-256. This text must be included in the discussion section.

Response: Thank you for the suggestion. As recommended, we have relocated this comparison to the Discussion section in the revised manuscript.

f) Lines 317-321. Panels of figure 4 should be labeled accordingly.

Response: We appreciate you catching this error. We have corrected the figure order; what was previously labeled as Figure 4F is now Figure 4E, and vice versa.

g) Line 379. Is supplemental figure 5 or 4?

Response: We appreciate you bringing this to our attention. It should be supplemental figure 4 in last version. In the revised manuscript, the virulence assays for the $\Delta dacA$ and $\Delta nagA$ mutants have been relocated to the supplemental data (now designated as Supplemental Figure 4). Consequently, it is now Supplemental Figure 5.

Re: Spectrum00122-25R3 (Transcription Factor RON1-Driven GlcNAc Catabolism Is Essential for Growth, Cell Wall Integrity, and Pathogenicity in *Aspergillus fumigatus*)

Dear Prof. Wenxia FANG:

Your manuscript has been accepted, and I am forwarding it to the ASM production staff for publication. Your paper will first be checked to make sure all elements meet the technical requirements. ASM staff will contact you if anything needs to be revised before copyediting and production can begin. Otherwise, you will be notified when your proofs are ready to be viewed.

Sincerely,
Miguel Penalva
Editor
Microbiology Spectrum